# DPsurv: Dual-Prototype Evidential Fusion for Uncertainty-Aware and Interpretable Whole Slide Image Survival Prediction

Yucheng Xing [1 2]   Ling Huang [1 3]   Jingying Ma [1]   Ruping Hong [4]   Jiangdong Qiu [4]   Pei Liu [5]   Kai He [1]
Huazhu Fu [6]   Mengling Feng [1]

## Abstract

Whole-slide images (WSIs) are widely used for cancer survival analysis because of their comprehensive histopathological information at both cellular and tissue levels, enabling quantitative, large-scale, and prognostically rich tumor feature analysis. However, most existing WSI survival analysis methods struggle with limited interpretability and often overlook predictive uncertainty in heterogeneous slide images. In this paper, we propose DPsurv, a dual-prototype whole-slide image evidential fusion network that outputs uncertainty-aware survival intervals, and enables interpretable survival results through patch prototype distribution assignment, component prototype evidence reasoning, and component-wise relative risk aggregation. Experiments on five publicly available datasets demonstrate strong discriminative performance and well-calibrated predictions, validating its effectiveness and reliability. The interpretation of survival results provides transparency at the feature, reasoning, and decision levels, thereby enhancing the trustworthiness and interpretability of DPsurv. Code is available at https://github.com/YuchengXing99/DPsurv.

## 1. Introduction

Survival analysis, which predicts survival probabilities and outcomes over time, is a critical task in oncology for guiding therapeutic decision-making and improving patient outcomes. As a direct reflection of tumor progression, whole-slide images (WSIs) have recently emerged as an essential source of data for survival prediction in computational pathology (Zhang et al., 2025). The major challenges in deriving reliable survival predictions from WSIs stem from their gigapixel scale and tissue heterogeneity (Xu et al., 2024). Failing to model and address these challenges can result in incomplete risk assessments, leading to suboptimal treatment planning and potentially compromised survival outcomes (Liu et al., 2025b; Shi et al., 2024b).

Prior work has largely focused on learning effective WSI representations for survival prediction, using strategies such as multiple instance learning (MIL), patch clustering (Claudio Quiros et al., 2024), or prototype-based representations (Song et al., 2024). Although these strategies manage to cope with high-resolution images, they insufficiently address tissue heterogeneity, which directly limits interpretability and erodes trust in the prediction. Clinically, distinct tissue components such as tumor epithelium, stroma, and necrosis each carry independent prognostic value, and even within the same component, different morphological patterns can imply different survival outcomes (Travis et al., 2011). Therefore, detecting and interpreting the subtle or ambiguous regions of WSIs that often carry decisive prognostic value is critical for reliable survival prediction.

Another important but overlooked consequence of tissue heterogeneity is the reliability of survival results. The inherent heterogeneity of WSIs and the presence of incomplete event labels (censoring) introduce uncertainty in survival outcomes (Gomes et al., 2021; Davidov et al., 2025). Conventional methods output point-level survival estimates without conveying the appropriate uncertainty, leading to unreliable treatment suggestions (Dolezal et al., 2022). Moreover, uncertainty research has been primarily focused on classification models for discrete output (Abdar et al., 2021; Yufei et al., 2022; Gal & Ghahramani, 2016; Lakshminarayanan et al., 2017; Jiang et al., 2025), while the study of uncertainty modeling for survival tasks remains limited.

Inspired by these insights, we propose DPsurv, a Dual-Prototype Evidential Fusion Network for interpretable and reliable WSI survival prediction. It encodes WSIs into deep

---

[1]National University of Singapore [2]National University of Singapore Guangzhou Research Translation and Innovation Institute [3]Imperial College London [4]Peking Union Medical College Hospital, Chinese Academy of Medical Sciences & Peking Union Medical College [5]Hunan University [6]Institute of High Performance Computing, Agency for Science, Technology and Research (A*STAR). Correspondence to: Ling Huang <iweisskohl@gmail.com>.

*Proceedings of the 43ʳᵈ International Conference on Machine Learning*, Seoul, South Korea. PMLR 306, 2026. Copyright 2026 by the author(s).

component embeddings using a patch prototype-guided Gaussian mixture model (GMM) and maps them into an evidence space with component prototype-based Gaussian random fuzzy numbers (GRFNs). The component-level evidence is then aggregated into relative survival risk to generate survival predictions with lower and upper bounds. The contributions are as follows:

- **Dual-Prototype Evidential Fusion for Reliable WSI Survival Prediction.** We introduce a Dual-Prototype Evidential Fusion Network that improves WSI prediction reliability by linking heterogeneous tissues to survival predictions with feature embedding, evidence modeling and fusion, and providing prediction intervals with aleatoric and epistemic uncertainty.

- **End-to-End Interpretable Survival Analysis.** The design of DPsurv enables end-to-end interpretability in survival analysis, tracing pathways from deep embeddings to survival evidence and ultimately to component-level relative risk, bringing transparency at feature, reasoning, and decision levels, thereby enhancing model trustworthiness.

- **Extensive Validation and State-of-the-Art Performance.** We conduct extensive experiments and evaluations to assess the discriminative ability and calibration performance of DPsurv, establishing a new state-of-the-art performance on several benchmarks.

## 2. Related work

Deep neural networks have advanced WSI survival analysis (Dimitriou et al., 2019), largely through weakly-supervised or unsupervised representation learning (Song et al., 2024). Although these methods address the gigapixel scale, tissue heterogeneity remains underexplored. Interpretability work in weak supervision relies on attention heatmaps (Shao et al., 2021; Xiang & Zhang, 2023), which offer relative importance but limited clinical meaning for risk assessment. Unsupervised approaches use prototypes or clustering to explain aggregation (Vu et al., 2023), yet provide only coarse feature-level insight. Overall, end-to-end interpretability across feature modeling, reasoning, and decision stages is still lacking. Uncertainty modeling has focused mainly on classification (Abdar et al., 2021) via Bayesian methods (Yufei et al., 2022), Monte Carlo dropout (Gal & Ghahramani, 2016), ensembling (Lakshminarayanan et al., 2017), or Subjective Logic (Jiang et al., 2025). In contrast, uncertainty in survival analysis is less studied. Recent GRFNs formulations enable explicit aleatoric and epistemic uncertainty in regression (Denœux, 2021; 2023b) and show promise for noisy, censored survival data (Huang et al., 2024b; 2025). Further details are in the Appendix B.

## 3. Method

### 3.1. Method Overview

The goal of DPsurv is to provide end-to-end interpretability and uncertainty quantification for reliable WSI survival prediction. Achieving these two objectives simultaneously is non-trivial, as it requires the model to remain transparent not only at the feature level, but also at the reasoning and decision levels.

To this end, DPsurv is formulated as a survival-evidence pipeline that progressively enhances modeling interpretability and reliability. Specifically, (i) **Preliminaries** introduce GRFNs, which jointly model aleatoric and epistemic uncertainty and yield interval-valued survival predictions through belief and plausibility functions; (ii) **Deep Slide Component Embedding** enables an interpretable representation of WSIs into morphological components at the feature level; (iii) **Component Evidence Modeling** maps these components to survival risk evidence at the reasoning level; and (iv) **Component Evidence Mixture** aggregates component-wise evidence at the evidence space, preserving uncertainty and enabling decision-level interpretability.

Intuitively, DPsurv treats a WSI as a collection of heterogeneous tissue patterns that contribute unequally to patient prognosis. Instead of collapsing all regions into a single deterministic representation, each morphological component is modeled as survival evidence with its own predictive uncertainty. Components associated with aggressive tumor morphology tend to provide stronger and more confident prognostic evidence, whereas stromal, necrotic, or ambiguous regions naturally produce weaker or more uncertain evidence. By aggregating component-wise GRFNs in the evidence space, DPsurv preserves these uncertainty characteristics throughout the prediction process, yielding an interpretable slide-level survival estimate together with calibrated uncertainty. For clarity, the key notations and symbols used throughout DPsurv are summarized in Appendix A.

### 3.2. Preliminaries

**Gaussian Random Fuzzy Numbers.** GRFNs (Denœux, 2023b) are adopted to explicitly model both aleatoric and epistemic uncertainty in survival results.

**Definition 3.1.** A GRFN represents uncertainty over a real-valued quantity by combining a Gaussian random variable with a fuzzy membership function. Formally, a GRFN is denoted as $\tilde{Y} \sim \tilde{\mathcal{N}}(\mu, \sigma^2, h)$, where $\mu$ is the location parameter, $\sigma^2$ captures aleatoric uncertainty, and $h \in [0, +\infty)$ quantifies epistemic uncertainty associated with limited or conflicting evidence.

Given a GRFN $\tilde{Y}$, uncertainty is characterized by a pair

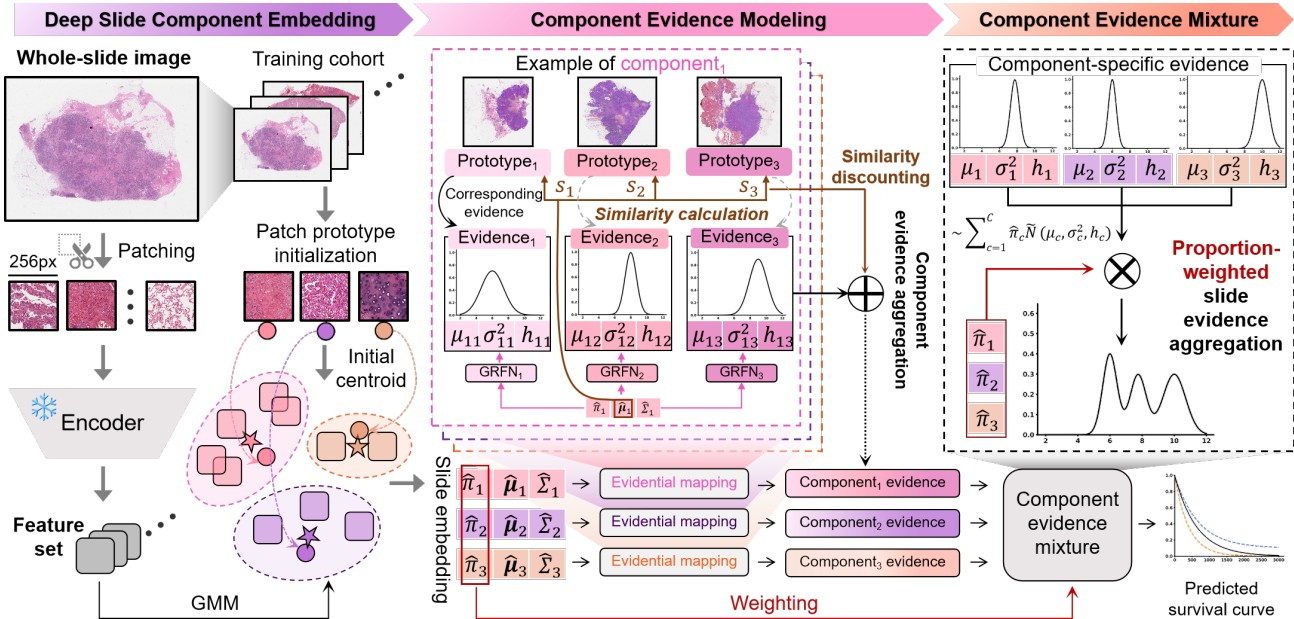

*Figure 1.* **Overview of the DPsurv framework** Deep Slide Component Embedding encodes WSI into deep feature embeddings with patch prototypes; Component Evidence Modeling maps deep embeddings into evidence through component prototypes, and Component Evidence Mixture aggregates component evidence into transformed survival functions, illustrated by the Plausibility (blue dashed line) and Belief (orange dashed line) survival curves.

of belief and plausibility functions, $Bel_{\tilde{Y}}(\cdot)$ and $Pl_{\tilde{Y}}(\cdot)$, which define degrees of belief and plausibility. For any interval $[x, y]$, both functions admit closed-form expressions that depend on the GRFN parameters $(\mu, \sigma, h)$. The full formulations and derivations are provided in Appendix C.

**Prediction Intervals under GRFNs.** With GRFNs, two types of prediction intervals can be constructed. An $\alpha$-level belief prediction interval (BPI) is defined as $\mu \pm v$ such that $Bel_{\tilde{Y}}([\mu - v, \mu + v]) = \alpha$, thereby explicitly accounting for epistemic uncertainty. In contrast, an $\alpha$-level probabilistic prediction interval (PPI) is given by $\mu \pm \Phi^{-1}\left(\frac{1+\alpha}{2}\right)\sigma$, which depends solely on Gaussian variance and ignores epistemic uncertainty. As a result, BPIs naturally widen when evidence is insufficient, making them particularly suitable for conservative uncertainty-aware prediction.

**From GRFNs to Survival Prediction.** Adopting GRFNs for survival analysis is straightforward by modeling survival time using a logarithmic transformation.

**Proposition 3.2.** *Let $T \in (0, \infty)$ denote the survival time and $Y = \log T$ be its logarithmic transformation, which maps the positive domain onto the real line. Under a GRFN model $\tilde{Y}$, the true survival function $S(t) = \mathbb{P}(T > t)$ is bounded by the belief and plausibility measures as*

$$Bel_{\tilde{Y}}((\log t, \infty)) \leq S(t) \leq Pl_{\tilde{Y}}((\log t, \infty)). \quad (1)$$

Proof is given in Appendix D.1.

Proposition 3.2 provides interval-valued survival predictions

that naturally account for both aleatoric and epistemic uncertainty, enabling conservative and well-calibrated survival estimation under limited or heterogeneous evidence.

### 3.3. Deep Slide Component Embedding

To ensure interpretability at the slide feature representation stage, we model each WSI as a mixture of latent tissue components using patch prototypes and a GMM (Dempster et al., 1977; Kim, 2022). A WSI foundation model is used to map WSIs into patch-level embeddings, with each WSI for subject $i$ being segmented into non-overlapping patches $\mathbf{X}^i = \{\mathbf{x}_1^i, \ldots, \mathbf{x}_{N_i}^i\}$, $\mathbf{x}_n^i \in \mathbb{R}^{W \times H \times 3}$. It should be noted that DPsurv is not limited to foundation models and can be applied to any state-of-the-art feature extraction model. Consequently, a set of patch embeddings for WSI is represented by $\mathbf{Z}^i = \{\mathbf{z}_1^i, \ldots, \mathbf{z}_{N_i}^i\}$ with $\mathbf{z}_n^i = f_{\text{enc}}(\mathbf{x}_n^i) \in \mathbb{R}^d$, $f_{\text{enc}}(\cdot)$ is the foundation model. However, these high-dimensional patch embeddings make survival analysis challenging. Following PANTHER (Song et al., 2024), we map the high-dimensional representation $\mathbf{Z}^i \in \mathbb{R}^{N_i \times d}$ into low-dimensional embedding $\mathbf{z}_{\text{WSI}}^i \in \mathbb{R}^{C \times (2d+1)}$ while preserving essential morphological information using patch prototypes:

$$\mathbf{z}_{\text{WSI}}^i = \left[\sum_{n=1}^{N_i} \phi^i(\mathbf{z}_n^i, \mathbf{v}_1), \ldots, \sum_{n=1}^{N_i} \phi^i(\mathbf{z}_n^i, \mathbf{v}_C)\right], \quad (2)$$

where $\mathbf{v}_c \in \mathbb{R}^d$ is the patch prototype, and $\phi^i(\cdot, \cdot)$ is a similarity-based function that maps a patch embedding-

prototype pair into a post-aggregation component embedding. Specifically, the similarities are used for soft assignment rather than feature pooling. Patch prototypes are learned globally across the training cohort and are shared by all WSIs. GMM is then used to estimate $\phi^i(\cdot, \cdot)$ with the assumption that the patch embedding $\mathbf{z}_n^i$ is generated from a weighted sum of its conditional densities under each patch-prototype-aligned Gaussian component:

$$
\begin{aligned}
p(\mathbf{z}_n^i; \theta^i) &= \sum_{c=1}^{C} p(c_n^i = c; \theta^i) \cdot p(\mathbf{z}_n^i | c_n^i = c; \theta^i) \\
&= \sum_{c=1}^{C} \pi_c^i \cdot \mathcal{N}(\mathbf{z}_n^i; \boldsymbol{\eta}_c^i, \Sigma_c^i), \quad \text{s.t.} \sum_{c=1}^{C} \pi_c^i = 1,
\end{aligned}
\tag{3}
$$

where $\theta^i = \left\{ \pi_c^i, \boldsymbol{\eta}_c^i, \Sigma_c^i \right\}_{c=1}^{C}$ denote the GMM parameters (Appendix E), $\pi_c^i$ is the prior probability, $\mathbf{z}_n^i$ originates from the $c$-th Gaussian component, $\boldsymbol{\eta}_c^i, \Sigma_c^i$ represent a morphological prototype and its variations for the $c$-th Gaussian component. Finally, WSI $i$ is represented by $\mathbf{z}_{\text{WSI}}^i \in \mathbb{R}^{C \times (2d+1)}$

$$
\mathbf{z}_{\text{WSI}}^i = [\, \hat{\pi}_1^i, \hat{\boldsymbol{\eta}}_1^i, \hat{\Sigma}_1^i, \ldots, \hat{\pi}_C^i, \hat{\boldsymbol{\eta}}_C^i, \hat{\Sigma}_C^i \,].
\tag{4}
$$

**Embedding interpretation with assignment map.**

*Remark* 3.3. In the GMM framework, each embedding $\mathbf{z}_n^i$ is probabilistically associated with a set of prototypes, where the posterior responsibility of component $c$ is determined by the mixture weight $\pi_c^i$ and its Gaussian likelihood. Each embedding is then assigned to the patch prototype $\mathbf{v}_{c_n^*}$ with the highest posterior responsibility, formally defined as

$$
c_n^* = \arg \max_{c \in \{1, \ldots, C\}} \frac{\pi_c^i \mathcal{N}(\mathbf{z}_n^i; \boldsymbol{\eta}_c^i, \Sigma_c^i)}{\sum_{k=1}^{C} \pi_k^i \mathcal{N}(\mathbf{z}_n^i; \boldsymbol{\eta}_k^i, \Sigma_k^i)}.
\tag{5}
$$

According to Remark 3.3, the resulting assignment map reveals distinct morphological patterns across the WSI. Projecting these assignments back onto the WSI yields a patch prototype assignment map that highlights the spatial interpretable distribution of pathology-related visual concepts.

### 3.4. Component Evidence Modeling

Slide-level features summarize overall tissue makeup but do not show how specific morphological components drive survival risk or uncertainty. We therefore model evidence at the component level for finer-grained, interpretable reasoning. Specifically, each component is associated with a set of component prototypes that act as local risk experts for particular patterns, letting the model assign risk and uncertainty to individual components instead of generating a single global score. Inspired by Huang et al. (2025), we map deep component embeddings $\mathbf{z}_{\text{WSI}}$ into evidence space via component prototypes using GRFNs.

Let $\boldsymbol{p}_{c,1}, \ldots, \boldsymbol{p}_{c,K} \in \mathbb{R}^d$ denote $K$ component prototype vectors for the $c^{\text{th}}$ Gaussian component $\mathbf{z}_{\text{WSI-c}} = [\hat{\pi}_c, \hat{\boldsymbol{\eta}}_c, \hat{\Sigma}_c]$. For $\mathbf{z}_{\text{WSI-c}}$, the evidence provided by component prototype $\boldsymbol{p}_{c,k}$ is encoded in a GRFN

$$
\tilde{Y}_{c,k} \sim \tilde{\mathcal{N}}(\mu_{c,k}, \sigma_{c,k}^2, h_{c,k}),
\tag{6}
$$

where $\sigma_{c,k}^2$ and $h_{c,k}$ denote the variance and precision of prototype $\boldsymbol{p}_{c,k}$, and the mean is parameterized as $\mu_{c,k} = \boldsymbol{\beta}_{c,k}^\top \mathbf{z}_{\text{WSI-c}} + \beta_{c,k0}$, with $\boldsymbol{\beta}_{c,k} \in \mathbb{R}^{2d+1}$ the coefficient vector and $\beta_{c,k0} \in \mathbb{R}$ a scalar.

The evidence of the $c^{\text{th}}$ Gaussian component $\tilde{Y}_c \sim \tilde{\mathcal{N}}(\mu_c, \sigma_c^2, h_c)$ is obtained by aggregating the evidence of its component prototypes $\{\tilde{Y}_{c,k}\}_{k=1}^{K}$ using the unnormalized product–intersection rule

$$
\mu_c = \frac{\sum_{k=1}^{K} s_{c,k} h_{c,k} \mu_{c,k}}{\sum_{k=1}^{K} s_{c,k} h_{c,k}}, \quad \sigma_c^2 = \frac{\sum_{k=1}^{K} s_{c,k}^2 h_{c,k}^2 \sigma_{c,k}^2}{\left( \sum_{k=1}^{K} s_{c,k} h_{c,k} \right)^2},
\tag{7}
$$

where $h_c = \sum_{k=1}^{K} s_{c,k} h_{c,k}$. Here $s_{c,k}$ is the discounting similarity between Gaussian component $\mathbf{z}_{\text{WSI-c}}$ and prototype $\boldsymbol{p}_{c,k}$ defined by

$$
s_{c,k} = \exp\left[-\gamma_{c,k}^2 d_{\cos}(\hat{\boldsymbol{\eta}}_c, \boldsymbol{p}_{c,k})\right],
\tag{8}
$$

with $d_{\cos}(\hat{\boldsymbol{\eta}}_c, \boldsymbol{p}_{c,k}) = \frac{1}{2} \left( 1 - \frac{\hat{\boldsymbol{\eta}}_c^\top \boldsymbol{p}_{c,k}}{\|\hat{\boldsymbol{\eta}}_c\| \|\boldsymbol{p}_{c,k}\|} \right)$ the cosine distance, and $\gamma_{c,k} > 0$ is a learnable positive scalar. This similarity-based discounting ensures that a component prototype contributes strong evidence only when it is well supported by the observed component embedding. Consequently, epistemic uncertainty increases naturally when no prototype is sufficiently similar, reflecting a lack of reliable prior knowledge rather than data noise. An illustrative example is provided in Appendix J.

**Survival evidence interpretation with component prototypes.** To understand the role of the component prototype in survival evidence modeling, we retrieve training samples from the same component with the highest cosine similarity to that prototype, thereby providing interpretable pathological characterizations.

*Remark* 3.4. For each of the $c^{\text{th}}$ Gaussian component and its component prototype $\boldsymbol{p}_{c,k}$, we assign $\exp(\mu_c)$ and $\exp(\mu_{c,k})$ as the most plausible survival times (PST), which serve as quantitative indicators of the associated risk evidence.

In summary, the survival evidence of a component is derived through a similarity-based aggregation of the risk evidence contributed by its component prototypes, leading to reasoning level transparency.

### 3.5. Component Evidence Mixture

Once the component-wise survival evidence is obtained in the form of GRFNs, we next aggregate them to produce a

slide-level survival prediction at the evidence space. We use the evidence mixture mechanism (Denœux, 2023a) to preserve uncertainty information and enable decision-level interpretability. Let $W$ be a random variable taking values in $\{1, \ldots, C\}$, we reformulate slide embedding as

$$\mathbf{z}_{\text{WSI}} = \sum_{c=1}^{C} \mathbf{1}_{\{W=c\}} \cdot [\hat{\boldsymbol{\eta}}_c, \hat{\Sigma}_c], \qquad (9)$$

where $P(W = c) = \hat{\pi}_c$ denotes the prior probability that a patch embedding belongs to the $c^{\text{th}}$ Gaussian component. Accordingly, $\tilde{Y}_c$ is a conditional GRFN given $W = c$ and the slide-level evidence is a mixture GRFN (m-GRFN, Appendix F), denoted by $\tilde{Y} \sim \sum_{c=1}^{C} \hat{\pi}_c \tilde{\mathcal{N}}(\mu_c, \sigma_c^2, h_c)$. Importantly, this aggregation operates at the evidence level rather than averaging point predictions, thereby preserving prediction uncertainty and enabling decision-level interpretability.

**Proposition 3.5.** *Let $\tilde{Y} \sim \sum_{c=1}^{C} \pi_c \tilde{Y}_c$ be a mixture of component-wise GRFNs. The belief and plausibility functions induced by $\tilde{Y}$ are given by convex combinations of those mixture components:*

$$Bel_{\tilde{Y}}([x, y]) = \sum_{c=1}^{C} \pi_c \, Bel_{\tilde{Y}_c}([x, y]),$$

$$Pl_{\tilde{Y}}([x, y]) = \sum_{c=1}^{C} \pi_c \, Pl_{\tilde{Y}_c}([x, y]). \qquad (10)$$

Proof is given in Appendix D.2.

Following Proposition 3.5 and letting $y \to \infty$, the degrees of belief and plausibility induced by m-GRFN $\tilde{Y}$, which correspond to the lower and upper bounds of the survival function, are given by:

$$Bel_{\tilde{Y}}([x, \infty)) = 1 - \Phi\left(\frac{x - \mu}{\sigma}\right) - pl_{\tilde{Y}}(x) \qquad (11a)$$

$$+ pl_{\tilde{Y}}(x) \, \Phi\left(\frac{x - \mu}{\sigma\sqrt{1 + h\sigma^2}}\right), \qquad (11b)$$

$$Pl_{\tilde{Y}}([x, \infty)) = 1 - \Phi\left(\frac{x - \mu}{\sigma}\right) \qquad (11c)$$

$$+ pl_{\tilde{Y}}(x) \, \Phi\left(\frac{x - \mu}{\sigma\sqrt{1 + h\sigma^2}}\right). \qquad (11d)$$

For each WSI $i$, the survival function at time $t$ is:

$$S_i(t) = \lambda \, Bel_{\tilde{Y}^i}([\log t, \infty)) + (1 - \lambda) \, Pl_{\tilde{Y}^i}([\log t, \infty)), \qquad (12)$$

where $\lambda \in [0, 1]$ controls the degree of conservativeness in decision, with larger values favoring belief-based (risk-averse) predictions and smaller values relying more on plausibility-based (risk-tolerant) estimates, yielding a flexible decision-risk-controlled model. Further details on the impact of $\lambda$ are provided in Appendix H.

**Survival prediction interpretation with Component-wise Relative Risk.** Given an m-GRFN, we introduce a relative risk measure to measure the importance of component-specific evidence to the final survival outcome.

*Remark* 3.6. In a GRFN, $\mu$ denotes the plausible log-survival time and is inversely related to risk. Accordingly, for a WSI, the component-wise relative risk is defined as

$$r_c = 1 - \frac{\mu_c - \min_j \mu_j}{\max_j \mu_j - \min_j \mu_j}, \quad c = 1, \ldots, C. \quad (13)$$

Remark 3.6 enables the visualization of relative risk distributions across WSIs, providing tissue-level interpretability.

### 3.6. Mixture evidential loss

We propose a mixture evidential loss for survival prediction under uncertainty, which integrates mixture-based evidence with the negative log-likelihood loss (Zadeh & Schmid, 2020). This formulation links uncertainty with survival probability while addressing censored–uncensored weighting. We first partition uncensored survival times in the training set into $B$ quantile-based bins, denoted as $b_j = [T_j, T_{j+1})$, such that each bin contains the same number of uncensored samples. We then calculate the negative log-likelihood of uncensored and all subjects, respectively, with

$$\ell_i^{\text{unc}} = -(1 - c_i) \sum_{j=1}^{B} \mathbf{1}_{\{y_i \in b_j\}} \log\big(S_i(T_j) - S_i(T_{j+1})\big), \tag{14a}$$

$$\ell_i = \ell_i^{\text{unc}} - c_i \sum_{j=1}^{B} \mathbf{1}_{\{y_i \in b_j\}} \log S_i(T_{j+1}), \tag{14b}$$

where $c_i$ is the censoring indicator. The mixture evidential loss is then defined as

$$\mathcal{L}_{\text{Mix}} = \frac{1}{N} \sum_{i=1}^{N} \Big[(1 - \alpha)\, \ell_i + \alpha\, \ell_i^{\text{unc}}\Big], \tag{15}$$

where $\alpha \in [0, 1]$ is a trade-off parameter that balances censored and uncensored likelihood contributions, thus controlling the robustness of the objective.

## 4. Experiments

### 4.1. Experimental setup

**Datasets.** Five cancers provided by TCGA are tested: Breast Invasive Carcinoma (BRCA), Bladder Urothelial Carcinoma (BLCA), Uterine Corpus Endometrial Carcinoma (UCEC), Kidney Renal Clear Cell Carcinoma (KIRC), and Lung Adenocarcinoma (LUAD) (Appendix G.1). Following (Song et al., 2024), we use 5-fold site-stratified cross-validation to minimize distribution differences between the

training and test sets (Howard et al., 2021), where within each fold, 15% of training data is held out as a validation set.

**Baselines.** Methods without uncertainty-awareness (UA) are: ABMIL (Ilse et al., 2018), TransMIL (Shao et al., 2021), DSMIL (Li et al., 2021), AttnMISL (Yao et al., 2020), ILRA (Xiang & Zhang, 2023) and PANTHER (Song et al., 2024). Methods with UA are: EVREG (Huang et al., 2025), UMSA (Jiang et al., 2025) and BayesMIL (Yufei et al., 2022). UNI2-h (Chen et al., 2024), pre-trained on a large-scale internal histology dataset, was used as the feature extractor for all comparison methods in this paper. Implementation details of all baseline methods are provided in Appendix G.2.

**Evaluation Metrics.** We use the Concordance index (C-index) (Harrell et al., 1982) to assess discrimination and use the integrated Brier score (IBS) and integrated binomial log-likelihood (IBLL) (Graf et al., 1999) to evaluate calibration (See Appendix G.3).

## 4.2. Survival Prediction Accuracy

As shown in Table 1, DPsurv achieves competitive and consistent discriminative performance across multiple cancer types. It attains the highest mean C-index across the five TCGA cohorts (0.704), ranking first on BRCA (0.720), BLCA (0.625), LUAD (0.667), UCEC (0.766), and KIRC (0.741). For cross-cancer evaluation, all C-index values obtained by DPsurv exceed 0.62, demonstrating consistent performance across diverse cancer types, indicating that the learned component prototypes capture generalizable tissue features from heterogeneous histologies and effectively mitigate the influence of redundant information during evidence aggregation.

## 4.3. End-to-End Interpretability Analysis

**Feature-level interpretability.** We visualize the assignment map together with the **patch prototype** distribution to examine the morphological phenotypes captured in WSIs (Remark 3.3). As shown in Figure 2A, the learned patch prototypes correspond to distinct tissue phenotypes, including tumor regions with different cellular densities, necrotic and inflammatory regions, as well as surrounding normal lung and stromal tissue. To evaluate the pathological relevance of the learned prototypes, two board-certified pathologists independently annotated the morphological identity of each prototype while blinded to the model assignments. The strong agreement between the pathological annotations and the learned prototype assignments suggests that DPsurv captures clinically meaningful tissue structure rather than arbitrary feature clusters.

**Reasoning-level interpretability.** Figure 2B visualizes how each component is mapped to survival evidence. For a target component, its risk evidence is obtained by aggregating evidence from component prototypes according to cosine similarity. Prototypes with higher similarity contribute more strongly to the final evidence, while dissimilar prototypes are naturally suppressed, making the reasoning process transparent and traceable. Pathologists confirmed that the learned component prototypes correspond to distinct phenotypic subtypes, including solid sheets of tumor cells with minimal stroma, solid–acinar patterns with stromal plasma cell infiltration, and acinar–cribriform growth with prominent lymphoid infiltration. Necrosis and mitotic activity are most prominent in component prototype-1, less evident in prototype-2, and weakest in prototype-3, consistent with the relative risk reflected by their predicted PST values (Remark 3.4). Intuitively, PST represents the most plausible survival time associated with a prototype, while BPI characterizes the uncertainty of that estimate. Wider BPIs indicate greater epistemic uncertainty, typically arising from limited or conflicting evidence. Aggregating evidence across multiple component prototypes produces more stable survival estimates and narrower BPIs, as consistent evidence reduces uncertainty in the fused prediction.

**Decision-level interpretability.** Figure 2C illustrates how survival decisions can be interpreted within the **component evidence mixture**. By linking components to their corresponding risk evidence, DPsurv produces a spatial risk distribution over the WSI (Remark 3.6), allowing different tissue regions to be associated with distinct risk levels. Tumor regions are consistently assigned the highest risk, which was further confirmed by pathologists. By further examining local ROIs, the model can directly associate localized morphological patterns with their predicted risk levels, enabling fine-grained risk attribution beyond slide-level predictions. This is particularly useful in heterogeneous tumor microenvironments, where identifying localized high-risk regions is often clinically important. Additional interpretability results and an expert evaluation by two board-certified pathologists are provided in Appendix I.

**Comparison with attention maps.** While attention heatmaps provide a form of interpretability by indicating which patches receive higher weights in the decision process, they remain limited to the feature level. In contrast, DPsurv provides a more structured and clinically meaningful interpretability framework, moving beyond patch-level weighting to deliver multi-level, pathology-informed, and quantitatively grounded explanations that better align with clinical reasoning and decision-making. We further discuss the potential utility of DPsurv in Appendix M.

## 4.4. Survival Uncertainty with Epistemic Modeling

Table 1 summarizes the uncertainty quantification performance in terms of both IBS and IBLL. Among the compared

*Table 1.* Main results from 5-fold cross-validation on five cancer datasets, together with the average performance across datasets. Best results are shown in **bold**, second-best results are underlined, and our method is color-coded

| | Methods | BRCA | | | BLCA | | | LUAD | | |
|---|---|---|---|---|---|---|---|---|---|---|
| | | C-index↑ | IBS↓ | IBLL↓ | C-index↑ | IBS↓ | IBLL↓ | C-index↑ | IBS↓ | IBLL↓ |
| ×UA | ABMIL | 0.687±0.06 | 0.752±0.14 | 3.019±1.16 | 0.557±0.04 | 0.535±0.09 | 1.540±0.34 | 0.611±0.11 | 0.627±0.19 | 1.938±0.70 |
| | TransMIL | 0.607±0.11 | 0.934±0.09 | 5.231±2.01 | 0.578±0.08 | 0.823±0.12 | 3.239±1.10 | 0.618±0.08 | 0.792±0.16 | 3.351±1.21 |
| | DSMIL | 0.652±0.04 | 0.694±0.16 | 2.208±0.54 | 0.583±0.03 | 0.417±0.10 | 1.080±0.26 | 0.619±0.10 | 0.473±0.11 | 1.227±0.28 |
| | AttnMISL | 0.654±0.06 | 0.801±0.15 | 3.563±1.63 | 0.545±0.05 | 0.499±0.12 | 1.537±0.44 | 0.593±0.12 | 0.465±0.17 | 1.314±0.54 |
| | ILRA | 0.665±0.05 | 0.932±0.03 | 4.491±1.09 | 0.552±0.07 | 0.865±0.07 | 3.999±1.02 | 0.578±0.07 | 0.846±0.17 | 4.654±0.74 |
| | PANTHER | 0.696±0.05 | 0.837±0.08 | 2.683±0.59 | 0.601±0.06 | 0.530±0.11 | 1.331±0.27 | 0.588±0.04 | 0.667±0.16 | 1.819±0.54 |
| ✓UA | EVREG | 0.585±0.03 | 0.665±0.01 | 8.467±0.51 | 0.587±0.01 | 0.461±0.00 | 5.469±0.06 | 0.565±0.02 | 0.483±0.00 | 5.964±0.18 |
| | UMSA | 0.640±0.08 | 0.730±0.11 | 2.336±0.51 | 0.573±0.07 | 0.501±0.10 | 1.381±0.35 | 0.634±0.10 | 0.569±0.17 | 1.663±0.59 |
| | BayesMIL | 0.707±0.05 | 0.721±0.06 | 2.024±0.31 | 0.602±0.06 | 0.414±0.08 | 1.058±0.19 | 0.622±0.11 | 0.461±0.08 | 1.180±0.18 |
| | DPsurv | **0.720**±0.03 | **0.199**±0.03 | **0.566**±0.08 | **0.625**±0.06 | **0.410**±0.12 | **0.855**±0.14 | **0.667**±0.08 | **0.381**±0.06 | **1.130**±0.24 |

| | Methods | UCEC | | | KIRC | | | Average | | |
|---|---|---|---|---|---|---|---|---|---|---|
| | | C-index↑ | IBS↓ | IBLL↓ | C-index↑ | IBS↓ | IBLL↓ | C-index↑ | IBS↓ | IBLL↓ |
| ×UA | ABMIL | 0.636±0.11 | 0.788±0.07 | 2.802±0.17 | 0.703±0.10 | 0.517±0.20 | 1.670±0.94 | 0.639 | 0.644 | 2.194 |
| | TransMIL | 0.698±0.07 | 0.936±0.05 | 4.590±0.93 | 0.725±0.08 | 0.755±0.13 | 3.313±1.09 | 0.645 | 0.848 | 3.945 |
| | DSMIL | 0.690±0.09 | 0.652±0.13 | 1.871±0.45 | 0.693±0.05 | 0.455±0.11 | 1.216±0.28 | 0.647 | 0.538 | 1.520 |
| | AttnMISL | 0.673±0.10 | 0.703±0.14 | 2.190±0.60 | 0.683±0.03 | 0.501±0.15 | 1.479±0.54 | 0.630 | 0.594 | 2.017 |
| | ILRA | 0.716±0.07 | 0.920±0.04 | 4.074±0.92 | 0.659±0.09 | 0.858±0.07 | 4.325±1.13 | 0.634 | 0.884 | 4.309 |
| | PANTHER | 0.709±0.06 | 0.866±0.10 | 3.045±0.86 | 0.683±0.09 | 0.701±0.15 | 1.926±0.48 | 0.655 | 0.720 | 2.161 |
| ✓UA | EVREG | 0.657±0.02 | 0.612±0.01 | 7.071±0.29 | 0.594±0.01 | 0.541±0.00 | 5.544±0.17 | 0.598 | 0.552 | 6.503 |
| | UMSA | 0.641±0.11 | 0.758±0.14 | 2.717±1.08 | 0.693±0.08 | 0.506±0.17 | 1.524±0.57 | 0.636 | 0.613 | 1.924 |
| | BayesMIL | 0.736±0.10 | 0.679±0.12 | 1.941±0.47 | 0.703±0.05 | 0.434±0.15 | 1.171±0.41 | 0.674 | 0.542 | 1.475 |
| | DPsurv | **0.766**±0.05 | **0.251**±0.08 | **0.692**±0.19 | **0.741**±0.08 | **0.310**±0.11 | **0.879**±0.28 | **0.704** | **0.310** | **0.824** |

methods, DPsurv achieves the lowest mean IBS (0.310) across the five TCGA cohorts, indicating accurate and well-calibrated survival probability estimates. In addition, DPsurv attains the lowest mean IBLL (0.824), reflecting improved calibration of the predicted survival distributions. Together, these results show that DPsurv not only discriminates between high and low risk patients, but also provides reliable survival estimates with well-calibrated uncertainty across diverse cancer types, highlighting the role of epistemic uncertainty modeling in improving calibration ability for survival prediction.

We further examine the calibration behavior of the two categories of methods. Excluding EVREG, whose evidential regression is not tailored to high-dimensional WSI features, the remaining uncertainty-aware (UA) methods generally attain lower IBS and IBLL values than most non-UA baselines, indicating that explicitly modeling predictive uncertainty generally improves calibration. Non-UA methods, which optimize primarily for discrimination, tend to show more variable calibration across cohorts. See Appendix K for more UA approach comparison.

We discuss two observations from calibration plots in Figure 3. **(1) BPIs improve calibration.** A survival model is considered well calibrated when the empirical coverage of its prediction intervals matches the nominal confidence level, corresponding to curves lying along the diagonal. Compared with PPIs, which tend to fall below the diagonal, BPIs con-

sistently lie closer to the diagonal across all datasets. This indicates that incorporating epistemic uncertainty leads to improved coverage calibration in survival prediction. **(2) BPIs mitigate overconfident risk estimation.** In most cases, BPI curves lie slightly above the diagonal, reflecting a conservative coverage behavior. This suggests that epistemic uncertainty helps reduce the risk of underestimating survival risk when model confidence is limited, thereby illustrating the practical effect of epistemic uncertainty in regulating model confidence under limited information.

### 4.5. Ablation Study and Sensitivity Analysis

**Ablation study.** For a controlled comparison, all ablation variants are evaluated under a fixed configuration without early stopping of K. The ablation results in Table 2 show that each module plays a distinct role in the overall framework. GMM-based slide representations already provide strong performance across all evaluation metrics. Incorporating GRFNs further leads to substantial improvements in calibration, as evidenced by marked reductions in both IBS and IBLL. Notably, CP and EM are designed to work in concert rather than independently: CP decomposes the slide-level prediction into component-level evidence, but without EM, these components are aggregated uniformly without reflecting their actual prevalence in the slide, introducing noise for components with low or zero presence; EM addresses this by re-weighting component evidence according to the

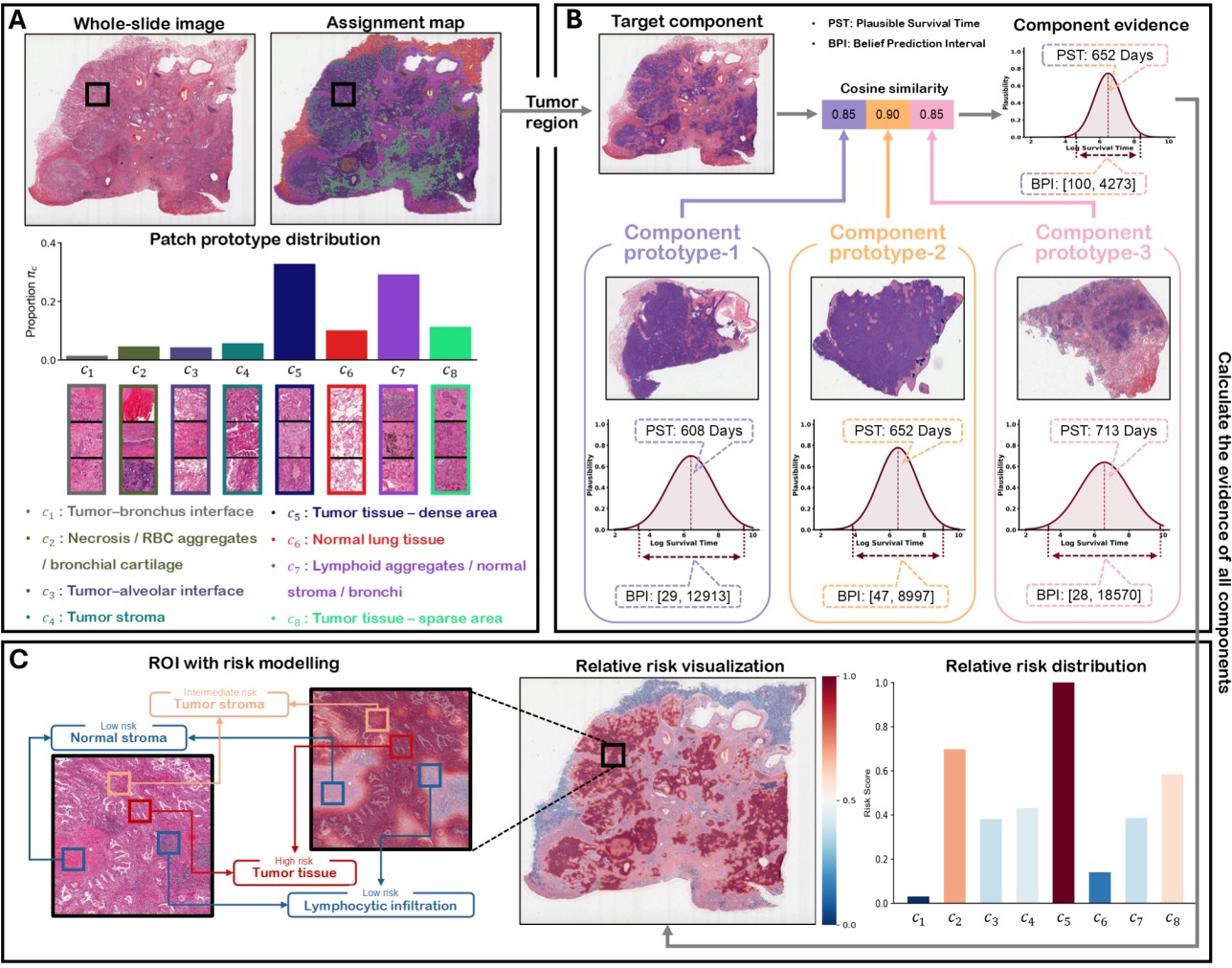

*Figure 2.* **Interpretation of the DPsurv in WSI survival prediction** (A) Visualization of the assignment map, prototype distribution, and morphological annotations provided by board-certified pathologists. (B) Visualization of component prototypes and the reasoning process for component evidence modeling. (C) Decision-making with component-wise relative risk and its distribution over the WSI and region of interest (ROI). Post-hoc annotation by pathologists is used solely to validate and interpret the semantic meaning of the learned prototypes, and is not a requirement for the model's interpretability.

*Table 2.* Ablation study of DPsurv on five TCGA cohorts, reporting mean performance across datasets. **FM**: Foundation Model used for patch-level feature extraction; **GMM**: deep slide component embedding with Gaussian mixture models; **GRFN**: Gaussian Random Fuzzy Number framework; **CP**: Component Prototype-based evidence modeling; **EM**: Component Evidence Mixture for aggregating component-wise survival evidence. Best results are shown in **bold**, second-best results are underlined.

| FM | GMM | GRFN | CP | EM | C-index | IBS | IBLL |
|----|-----|------|-----|-----|---------|------|------|
| ✓ | ✗ | ✗ | ✗ | ✗ | 0.621 | 0.732 | 3.223 |
| ✓ | ✓ | ✗ | ✗ | ✗ | 0.658 | 0.585 | 1.574 |
| ✓ | ✓ | ✓ | ✗ | ✗ | 0.663 | 0.331 | 0.894 |
| ✓ | ✓ | ✓ | ✓ | ✗ | 0.538 | 0.300 | 0.953 |
| ✓ | ✓ | ✓ | ✗ | ✓ | 0.645 | 0.220 | 0.720 |
| ✓ | ✓ | ✓ | ✓ | ✓ | **0.687** | **0.209** | **0.604** |

tissue composition prior $\hat{\pi}_c$, which explains the C-index drop observed when CP is used without EM. Together, CP and EM jointly enhance interpretability through explicit attribution of risk evidence to individual tissue components, while further improving calibration and preserving discriminative performance. In summary, the full model achieves the strongest discriminative performance while maintaining well-calibrated uncertainty estimates.

**Sensitivity analysis.** Appendix H reports the sensitivity of DPsurv to the two prototype-related hyperparameters: the number of patch prototypes $C$ and the number of component prototypes $K$ (fixed at specified values rather than selected by early stopping). Performance remains stable across a broad range of settings for both parameters. Specifically, on KIRC, varying $K$ from 2 to 6 and $C$ from 8 to

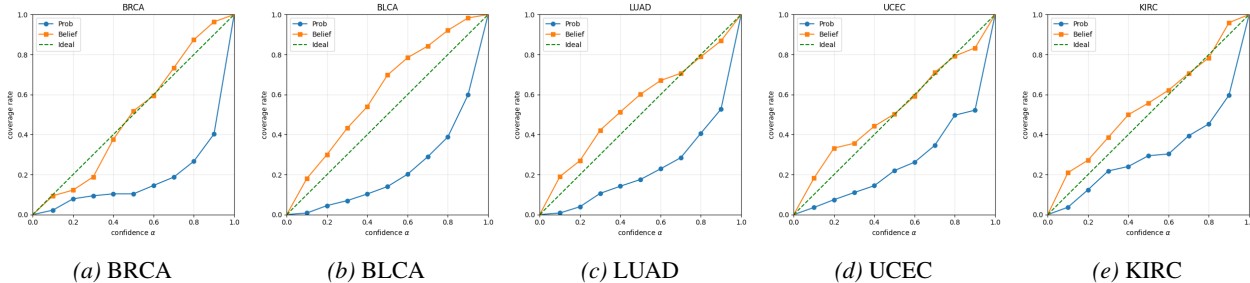

| *(a)* BRCA | *(b)* BLCA | *(c)* LUAD | *(d)* UCEC | *(e)* KIRC |
|---|---|---|---|---|

*Figure 3.* Calibration plots of DPsurv across five TCGA datasets, showing the proportion of $\alpha$-level BPIs and PPIs that contain the uncensored survival times, for $\alpha \in \{0.1, \ldots, 0.9\}$.

32 yields C-index values within 0.712–0.732 and 0.719–0.732, respectively; on UCEC, the corresponding ranges are 0.715–0.735 and 0.722–0.740, confirming robustness to prototype configuration choices. For the remaining hyperparameters, DPsurv also shows stable performance across a broad range of settings. In particular, the parameter $\lambda$ controls the trade-off between discrimination and calibration: larger values emphasize belief-based predictions, leading to more conservative and better-calibrated survival estimates. The parameter $\alpha$ balances the contribution of censored samples in the loss function; a larger $\alpha$ places greater emphasis on uncensored observations.

### 4.6. Feature Extractor Robustness

To assess robustness to the choice of feature extractor, we additionally evaluate DPsurv on UCEC using CONCH (Lu et al., 2023), a vision–language pathology foundation model whose pretraining strategy differs fundamentally from UNI2-h. As shown in Table 3, DPsurv achieves the best performance across all three metrics, indicating that its effectiveness does not depend on the geometric structure of a specific feature space and that the GMM-based decomposition and evidential fusion pipeline generalizes across feature extractors with different representational properties.

*Table 3.* Results on UCEC using CONCH as the feature extractor.

| Method | C-index↑ | IBS↓ | IBLL↓ |
|---|---|---|---|
| ABMIL | 0.757±0.05 | 0.695±0.15 | 2.244±0.74 |
| TransMIL | 0.749±0.06 | 0.799±0.16 | 2.808±0.87 |
| PANTHER | 0.732±0.11 | 0.791±0.15 | 2.625±0.87 |
| BayesMIL | 0.759±0.06 | 0.666±0.14 | 1.926±0.52 |
| **DPsurv** | **0.780**±0.10 | **0.239**±0.12 | **0.673**±0.31 |

### 4.7. Computational Cost

We evaluate the computational cost of DPsurv against simpler MIL models on BLCA (437 WSIs) and BRCA (1,111 WSIs), the smallest and largest cohorts in our study, cover-

ing both low- and high-volume settings. As shown in Figure 4, the overhead of DPsurv remains acceptable, largely because the GMM aggregation compresses each WSI into a compact set of mixture features and keeps the downstream evidential model lightweight. The similar runtime for BLCA and BRCA further suggests that DPsurv scales well to larger cohorts and higher-resolution WSIs.

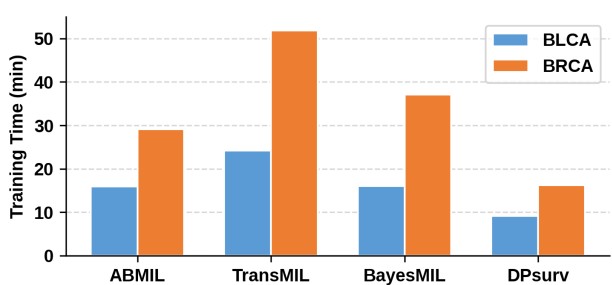

*Figure 4.* Training time comparison (in minutes) across representative MIL methods on the BLCA (437 WSIs) and BRCA (1,111 WSIs) cohorts, measured on a single NVIDIA A100 (80GB) under the default training configuration.

## 5. Conclusion

We propose DPsurv, a Dual-Prototype Evidential Fusion Network for reliable WSI survival prediction. To enhance interpretability and uncertainty awareness, our framework integrates three key parts: i) patch prototype–based GMMs to derive slide-level component embeddings, ii) component prototype–based evidential modeling to map component embeddings into component-wise evidence, and iii) evidence mixture mechanism to fuse component-wise evidence into final predictive evidence. DPsurv achieves state-of-the-art discriminative and well-calibrated performance across five cancer datasets. Additionally, the model provides multi-level transparency and explicit uncertainty quantification, offering a novel perspective for interpretable and uncertainty-aware survival prediction in computational pathology.

## Acknowledgements

This research is supported by the National Medical Research Council (NMRC) Healthy and Meaningful Longevity – Cognition Grant (NICCOG2024-0028) and the AI for Public Health Program in Saw Swee Hock School of Public Health, National University of Singapore. This work was also supported by the Dame Julia Higgins Postdoc Collaborative Research Fund and the NUS-Guangzhou Research Translation and Innovation Institute Scholarship.

## Impact Statement

This paper presents work whose goal is to advance the field of Machine Learning. There are many potential societal consequences of our work, none of which we feel must be specifically highlighted here.

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

# Appendix

## A. Notation and Terminology

Table 4 summarizes the key terms and symbols used in DPsurv.

*Table 4.* Summary of key terms and their roles in DPsurv.

| Term | Definition | Use / Role in DPsurv | Where used |
|---|---|---|---|
| Patch prototype ($\mathbf{v}_c$) | Global centroid in patch embedding space, learned via K-means. Shared across all WSIs. There are $C$ of them. | Used to partition patch embeddings into coarse morphological groups, such as tumor, stroma, or necrosis-like tissue patterns. They provide the basis for WSI decomposition. | Sec. 3.3 (Eq. 2–5) |
| Component | A Gaussian mixture component aligned to a patch prototype. Each WSI has $C$ components with parameters $(\hat{\pi}_c, \hat{\boldsymbol{\eta}}_c, \hat{\Sigma}_c)$. | Represents one morphology-aware tissue component in the WSI, modeled as a Gaussian distribution in feature space. It summarizes the prevalence, center, and variation of one tissue pattern. | Sec. 3.3 (Eq. 3–4) |
| Component prototype ($p_{c,k}$) | A learned vector in the slide embedding space. Each component $c$ has $K$ prototypes that serve as local risk experts. | Used to model finer-grained subtypes or risk patterns within one tissue component. They capture how different variants of the same tissue component relate to survival evidence. | Sec. 3.4 (Eq. 6–8) |
| Component evidence | The GRFN $\tilde{Y}_c$ produced by aggregating evidence from $K$ component prototypes for component $c$. | Represents the survival risk evidence associated with one tissue component, including both predicted survival tendency and its uncertainty. | Sec. 3.4 (Eq. 7) |
| Component evidence mixture | The m-GRFN that aggregates all $C$ component-level GRFNs into a slide-level prediction. | Mixtures the risk evidence from all tissue components into a final slide-level survival representation and prediction. | Sec. 3.5 (Eq. 10) |

## B. Detailed Related Work

**Learning approaches for WSI survival analysis.** Learning effective representations is a fundamental paradigm shared across domains (Ma et al., 2025; 2026; 2024). In computational pathology, WSI-based survival prediction research can be broadly grouped into weakly-supervised and unsupervised methods. Weakly-supervised approaches are predominantly based on MIL, where patch features are generated by pre-trained feature extractors and then aggregated into a slide-level representation and mapped to survival risk via a prediction head (Yao et al., 2020; Xiang & Zhang, 2023; Xing et al., 2026; Liu et al., 2026). Based on the aggregation approaches, MIL approaches can be categorized into cluster (Zhou et al., 2024; Liu et al., 2025a), attention (Yang et al., 2024; Jiang et al., 2024; Kapse et al., 2024), and graph-based (Zheng et al., 2024; Li et al., 2024; Bui et al., 2024). Studies also extend MIL into multiscale modeling to learn hierarchical representations (Chen et al., 2022; Li et al., 2022; Deng et al., 2024). Unsupervised representation learning aims to construct explicit slide-level representations that preserve morphological heterogeneity in an unsupervised manner with strategies such as patch embedding clustering (Zaheer et al., 2017; Vu et al., 2023; Yu et al., 2023; Claudio Quiros et al., 2024), prototype learning (Mialon et al., 2020; Xu & Chen, 2023) and compact morphological prototype learning (Song et al., 2024).

**Uncertainty and interpretability in WSI survival analysis.** Uncertainty studies in WSI survival analysis can be categorized into probabilistic and non-probabilistic methods (Huang et al., 2024a). Probabilistic approaches model uncertainty by relying on probability distributions. For example, Tang et al. (2025) estimates aleatoric uncertainty through a likelihood–based loss to provide fine-grained reliability scoring, Tang et al. (2023) injects aleatoric uncertainty into the Cox loss via a sample-dependent variance term, Yufei et al. (2022) applies a Bayesian formulation of attention weights to improve calibration. Non-probabilistic approaches model uncertainty without explicit probability distributions, often leveraging evidential or belief-based frameworks (Huang et al., 2024a). One representative method is Subjective Logic, which links evidential strength to the parameters of a Dirichlet distribution: Shi et al. (2024a) outputs Dirichlet evidence at the instance level and aggregates to bag-level predictions, Jiang et al. (2025) parameterizes survival predictions using Dirichlet evidence for calibration in multi-scale pathology–genomics fusion. Another direction employs evidential neural networks with

GRFNs to capture survival uncertainty in both unimodal and multimodal settings (Huang et al., 2024c; 2025). Specifically, Huang et al. (2024c) focuses on multimodal survival fusion across data sources, evaluating on image-clinical and clinical-genomic datasets without using histopathology WSIs as input. By contrast, DPsurv focuses on a single WSI and addresses a different question: how to decompose one WSI into morphology-aware components and perform uncertainty-aware and interpretable survival reasoning within that slide. The novelty of DPsurv lies in the combination of (i) prototype-guided WSI decomposition into morphology-aware components, (ii) component-wise evidential modeling where each component generates GRFN-based survival evidence, and (iii) evidence-level mixture that aggregates component evidence into interpretable slide-level predictions.

WSI survival interpretability studies have primarily been addressed within weakly-supervised MIL frameworks that use attention or pooling-related mechanisms to explain the aggregation process of slide embedding, such as local attention for patch-specific importance (Ilse et al., 2018), morphological prototypes (Yao et al., 2020), transformer modules for spatial correlations (Shao et al., 2021), multiscale embeddings with contrastive pretraining (Li et al., 2021), and low-rank attention for patch dependencies (Xiang & Zhang, 2023). Unsupervised approaches mainly focus on constructing morphology-associated slide representations for interpretation study through feature averaging, cluster counts, or optimal-transport/GMM-based prototypes (Zaheer et al., 2017; Mialon et al., 2020; Claudio Quiros et al., 2024; Yu et al., 2023; Vu et al., 2023; Song et al., 2024). However, these strategies often focus only on feature-level interpretability, thereby limiting model expressivity and transparency in the decision pathway from histological representation to survival outcomes.

## C. Additional Preliminaries

**Derivation of the contour function.** In Epistemic Random Fuzzy Sets (ERFS) theory, a GRFN is a random fuzzy subset of the real line defined by the membership function $\varphi(x; M, h) = \exp\left(-\frac{h}{2}(x - M)^2\right)$, whose mode $M$ is a Gaussian random variable with $M \sim \mathcal{N}(\mu, \sigma^2)$ (Denœux, 2023b). A GRFN can be represented by $\tilde{Y} \sim \tilde{\mathcal{N}}(\mu, \sigma^2, h)$, where $\mu$ is the location parameter, and $\sigma$ and $h \in [0, +\infty)$ represent the aleatoric and epistemic uncertainties, respectively.

Starting from the definition,

$$
\begin{aligned}
pl_{\tilde{Y}}(x) = \mathbb{E}_M\big[\varphi(x; M, h)\big] &= \int_{-\infty}^{+\infty} \varphi(x; m, h)\, \phi(m; \mu, \sigma)\, dm \\
&= \frac{1}{\sigma\sqrt{2\pi}} \int_{-\infty}^{+\infty} \exp\big(-\tfrac{h}{2}(x - m)^2\big) \exp\left(-\tfrac{(m-\mu)^2}{2\sigma^2}\right) dm.
\end{aligned}
\tag{16}
$$

Following Denœux (2023b), the integrand can be factorized as

$$
\exp\left(-\frac{(m - \mu_0)^2}{2\sigma_0^2}\right) \exp\left(-\frac{h(x - \mu)^2}{2(1 + h\sigma^2)}\right),
\tag{17}
$$

with

$$
\mu_0 = \frac{x\,h\sigma^2 + \mu}{1 + h\sigma^2}, \qquad \sigma_0^2 = \frac{\sigma^2}{1 + h\sigma^2}.
\tag{18}
$$

Evaluating the Gaussian integral then yields

$$
\begin{aligned}
pl_{\tilde{Y}}(x) &= \frac{1}{\sigma\sqrt{2\pi}} \exp\left(-\frac{h(x - \mu)^2}{2(1 + h\sigma^2)}\right) \int_{-\infty}^{+\infty} \exp\left(-\frac{(m - \mu_0)^2}{2\sigma_0^2}\right) dm \\
&= \frac{1}{\sqrt{1 + h\sigma^2}} \exp\left(-\frac{h(x - \mu)^2}{2(1 + h\sigma^2)}\right).
\end{aligned}
\tag{19}
$$

**Derivation of the plausibility function.** Assume $h > 0$. By the definition of plausibility over an interval,

$$
\begin{aligned}
Pl_{\tilde{Y}}([x,y]) &= P(M \leq x)\,\mathbb{E}\big[\varphi(x; M, h) \mid M \leq x\big] + P(x < M \leq y) \cdot 1 \\
&\quad + P(M > y)\,\mathbb{E}\big[\varphi(y; M, h) \mid M > y\big] \\
&= \Phi\big(\tfrac{x-\mu}{\sigma}\big)\,\mathbb{E}\big[\varphi(x; M, h) \mid M \leq x\big] + \Big(\Phi\big(\tfrac{y-\mu}{\sigma}\big) - \Phi\big(\tfrac{x-\mu}{\sigma}\big)\Big) \\
&\quad + \Big(1 - \Phi\big(\tfrac{y-\mu}{\sigma}\big)\Big)\mathbb{E}\big[\varphi(y; M, h) \mid M > y\big].
\end{aligned}
\tag{20}
$$

Since $M \mid M \leq x$ is a Gaussian truncated to $(-\infty, x]$ with probability density function

$$
f(m) = \frac{1}{\sigma\sqrt{2\pi}\,\Phi\big(\frac{x-\mu}{\sigma}\big)} \exp\Big(-\tfrac{(m-\mu)^2}{2\sigma^2}\Big)\,\mathbf{1}_{(-\infty, x]}(m),
\tag{21}
$$

we get

$$
\mathbb{E}\big[\varphi(x; M, h) \mid M \leq x\big] = \frac{1}{\sigma\sqrt{2\pi}\,\Phi\big(\frac{x-\mu}{\sigma}\big)} \int_{-\infty}^{x} \exp\big(-\tfrac{h}{2}(x-m)^2\big)\exp\Big(-\tfrac{(m-\mu)^2}{2\sigma^2}\Big)\,dm.
\tag{22}
$$

Direct evaluation provides a closed form in terms of $\Phi(\cdot)$ and the previously derived $pl_{\tilde{Y}}(x)$; an analogous computation applies to $\mathbb{E}\big[\varphi(y; M, h) \mid M > y\big]$. Substituting both expressions into (20) gives the stated formula:

$$
Pl_{\tilde{Y}}([x,y]) = \Phi\big(\tfrac{y-\mu}{\sigma}\big) - \Phi\big(\tfrac{x-\mu}{\sigma}\big) + pl_{\tilde{Y}}(x)\,\Phi\Big(\tfrac{x-\mu}{\sigma\sqrt{1+h\sigma^2}}\Big) + pl_{\tilde{Y}}(y)\Big[1 - \Phi\Big(\tfrac{y-\mu}{\sigma\sqrt{1+h\sigma^2}}\Big)\Big].
\tag{23}
$$

**Derivation of the belief function.** By duality between plausibility and belief,

$$
Bel_{\tilde{Y}}([x,y]) = 1 - Pl_{\tilde{Y}}\big((-\infty, x] \cup [y, +\infty)\big).
\tag{24}
$$

Using the same decomposition as for plausibility, we write

$$
\begin{aligned}
Pl_{\tilde{Y}}\big((-\infty, x] \cup [y, +\infty)\big) &= P(M \leq x) \cdot 1 \\
&\quad + P\big(x < M \leq \tfrac{x+y}{2}\big)\,\mathbb{E}\big[\varphi(x; M, h) \mid x < M \leq \tfrac{x+y}{2}\big] \\
&\quad + P\big(\tfrac{x+y}{2} < M \leq y\big)\,\mathbb{E}\big[\varphi(y; M, h) \mid \tfrac{x+y}{2} < M \leq y\big] \\
&\quad + P(M > y) \cdot 1 \\
&= \Phi\big(\tfrac{x-\mu}{\sigma}\big) \\
&\quad + \left[\Phi\Big(\tfrac{(x+y)/2-\mu}{\sigma}\Big) - \Phi\big(\tfrac{x-\mu}{\sigma}\big)\right]\mathbb{E}\big[\varphi(x; M, h) \mid x < M \leq \tfrac{x+y}{2}\big] \\
&\quad + \left[\Phi\big(\tfrac{y-\mu}{\sigma}\big) - \Phi\Big(\tfrac{(x+y)/2-\mu}{\sigma}\Big)\right]\mathbb{E}\big[\varphi(y; M, h) \mid \tfrac{x+y}{2} < M \leq y\big] \\
&\quad + 1 - \Phi\big(\tfrac{y-\mu}{\sigma}\big).
\end{aligned}
\tag{25}
$$

On $x < M \leq (x+y)/2$, $M$ follows a Gaussian truncated to $(x, (x+y)/2]$ with probability density function

$$
f(m) = \frac{1}{\sigma\sqrt{2\pi}\,\big[\Phi(\frac{(x+y)/2-\mu}{\sigma}) - \Phi(\frac{x-\mu}{\sigma})\big]} \exp\Big(-\tfrac{(m-\mu)^2}{2\sigma^2}\Big)\,\mathbf{1}_{(x,(x+y)/2]}(m).
\tag{26}
$$

Therefore,

$$\mathbb{E}\big[\varphi(x; M, h) \mid x < M \leq \tfrac{x+y}{2}\big] = \frac{\int_x^{(x+y)/2} \exp\big(-\tfrac{h}{2}(x-m)^2\big) \exp\big(-\tfrac{(m-\mu)^2}{2\sigma^2}\big)\, dm}{\sigma\sqrt{2\pi}\left[\Phi\big(\tfrac{(x+y)/2-\mu}{\sigma}\big) - \Phi\big(\tfrac{x-\mu}{\sigma}\big)\right]}$$

$$= \frac{\Phi\big(\tfrac{(x+y)/2-\mu+h\sigma^2(y-x)/2}{\sigma\sqrt{h\sigma^2+1}}\big) - \Phi\big(\tfrac{x-\mu}{\sigma\sqrt{h\sigma^2+1}}\big)}{\Phi\big(\tfrac{(x+y)/2-\mu}{\sigma}\big) - \Phi\big(\tfrac{x-\mu}{\sigma}\big)}\, pl_{\tilde{Y}}(x). \tag{27}$$

Similarly,

$$\mathbb{E}\big[\varphi(y; M, h) \mid \tfrac{x+y}{2} < M \leq y\big] = \frac{\Phi\big(\tfrac{y-\mu}{\sigma\sqrt{h\sigma^2+1}}\big) - \Phi\big(\tfrac{(x+y)/2-\mu-(y-x)h\sigma^2/2}{\sigma\sqrt{h\sigma^2+1}}\big)}{\Phi\big(\tfrac{y-\mu}{\sigma}\big) - \Phi\big(\tfrac{(x+y)/2-\mu}{\sigma}\big)}\, pl_{\tilde{Y}}(y). \tag{28}$$

Plugging these into (24) and (25) produces the closed-form expression of belief function:

$$Bel_{\tilde{Y}}([x,y]) = \Phi\big(\tfrac{y-\mu}{\sigma}\big) - \Phi\big(\tfrac{x-\mu}{\sigma}\big) - pl_{\tilde{Y}}(x)\left[\Phi\big(\tfrac{(x+y)/2-\mu+(y-x)h\sigma^2/2}{\sigma\sqrt{1+h\sigma^2}}\big) - \Phi\big(\tfrac{x-\mu}{\sigma\sqrt{1+h\sigma^2}}\big)\right]$$

$$+ pl_{\tilde{Y}}(y)\left[\Phi\big(\tfrac{(x+y)/2-\mu-(y-x)h\sigma^2/2}{\sigma\sqrt{1+h\sigma^2}}\big) - \Phi\big(\tfrac{y-\mu}{\sigma\sqrt{1+h\sigma^2}}\big)\right], \tag{29}$$

## D. Proof

### D.1. Proof of Proposition 3.2

*Proof.* Let $Y = \log T$ and $S(t) = \mathbb{P}(T > t) = \mathbb{P}(Y > \log t)$. Under the Dempster–Shafer interpretation of a GRFN $\tilde{Y}$, the belief and plausibility functions provide lower and upper bounds on the probability of any measurable event $A$, i.e.,

$$Bel_{\tilde{Y}}(A) \;\leq\; \mathbb{P}(Y \in A) \;\leq\; Pl_{\tilde{Y}}(A).$$

Choosing $A = (\log t, \infty)$ immediately yields

$$Bel_{\tilde{Y}}((\log t, \infty)) \;\leq\; S(t) \;\leq\; Pl_{\tilde{Y}}((\log t, \infty)),$$

which completes the proof. $\qquad\square$

### D.2. Proof of Proposition 3.5

*Proof.* By the definition of plausibility, $Pl_{\tilde{Y}}([x,y])$ can be expressed as

$$Pl_{\tilde{Y}}([x,y]) = \mathbb{E}_{M,W}\left[\sup_{u\in[x,y]} \varphi\left(u, M, \prod_{c=1}^C h_c^{I(W=c)}\right)\right]$$

$$= \mathbb{E}_W \mathbb{E}_{M|W}\left[\sup_{u\in[x,y]} \varphi\left(u, M, \prod_{c=1}^C h_c^{I(W=c)}\right)\right]$$

$$= \sum_{c=1}^C \pi_c\, \mathbb{E}_{M|W}\left[\sup_{u\in[x,y]} \varphi(u, M, h_c) \,\big|\, W = c\right] \tag{30}$$

$$= \sum_{c=1}^C \pi_c\, Pl_{\tilde{Y}_c}([x,y]).$$

Here, $Pl_{\tilde{Y}}([x,y])$ denotes the plausibility function of the mixture GRFN. By duality, the belief function $Bel_{\tilde{Y}}([x,y])$ can be derived in the same way, and is interpreted as the belief function associated with the same mixture GRFN. $\qquad\square$

## E. Gaussian Mixture Model Estimation Details

For completeness, we provide the detailed derivation of the parameter estimation for the Gaussian Mixture Model described in Section 3.3.

We clarify that $\phi^i(\cdot, \cdot) : \mathbb{R}^d \times \mathbb{R}^d \to \mathbb{R}^D$ takes as input the $d$-dimensional patch embedding $\mathbf{z}_n^i$ and a patch prototype $\mathbf{v}_c$, and computes a $D$-dimensional component-wise aggregated representation. In particular, $\phi^i(\cdot, \cdot)$ measures the similarity of each patch to the Gaussian components and uses it to weight the patch's contribution to the aggregated representation, producing a post-aggregation prototype-level embedding for downstream fusion. Notably, the parameters of the GMM are fixed and do not participate in the training of the subsequent evidential fusion network.

In our formulation, each Gaussian component contributes $(2d + 1)$ dimensions, consisting of the component mean ($d$ dimensions), the component variance ($d$ dimensions), and the mixture weight (one dimension). Hence, the component-wise representation has size $D = 2d + 1$. With $C$ components in the mixture, the final per-slide representation lies in $\mathbb{R}^{C \times (2d+1)}$.

Given the patch embeddings $\mathbf{Z}^i = \{\mathbf{z}_1^i, \dots, \mathbf{z}_{N_i}^i\}$ of WSI $i$, the likelihood of the Gaussian mixture model is

$$p(\mathbf{Z}^i; \theta^i) = \prod_{n=1}^{N_i} p(\mathbf{z}_n^i; \theta^i) = \prod_{n=1}^{N_i} \sum_{c=1}^{C} \pi_c^i \mathcal{N}(\mathbf{z}_n^i; \boldsymbol{\eta}_c^i, \Sigma_c^i), \tag{31}$$

where $\theta^i = \{\pi_c^i, \boldsymbol{\eta}_c^i, \Sigma_c^i\}_{c=1}^{C}$ are the parameters of the mixture.

The log-likelihood is

$$\ell(\theta^i) = \sum_{n=1}^{N_i} \log \left( \sum_{c=1}^{C} \pi_c^i \mathcal{N}(\mathbf{z}_n^i; \boldsymbol{\eta}_c^i, \Sigma_c^i) \right). \tag{32}$$

Direct maximization of $\ell(\theta^i)$ is intractable due to the summation inside the logarithm. Instead, we apply the Expectation-Maximization algorithm.

**E-step.** For each observation $\mathbf{z}_n^i$, the posterior probability that it belongs to component $c$ under parameters $\theta^{i(t)}$ is

$$p(c_n^i = c \mid \mathbf{z}_n^i; \theta^{i(t)}) = \frac{\pi_c^{i(t)} \mathcal{N}(\mathbf{z}_n^i; \boldsymbol{\eta}_c^{i(t)}, \Sigma_c^{i(t)})}{\sum_{c'=1}^{C} \pi_{c'}^{i(t)} \mathcal{N}(\mathbf{z}_n^i; \boldsymbol{\eta}_{c'}^{i(t)}, \Sigma_{c'}^{i(t)})}. \tag{33}$$

**M-step.** The parameters are updated by maximizing the expected complete-data log-likelihood:

$$\pi_c^{i(t+1)} = \frac{1}{N_i} \sum_{n=1}^{N_i} p(c_n^i = c \mid \mathbf{z}_n^i; \theta^{i(t)}), \tag{34}$$

$$\boldsymbol{\eta}_c^{i(t+1)} = \frac{\sum_{n=1}^{N_i} p(c_n^i = c \mid \mathbf{z}_n^i; \theta^{i(t)}) \, \mathbf{z}_n^i}{\sum_{n=1}^{N_i} p(c_n^i = c \mid \mathbf{z}_n^i; \theta^{i(t)})}, \tag{35}$$

$$\Sigma_c^{i(t+1)} = \frac{\sum_{n=1}^{N_i} p(c_n^i = c \mid \mathbf{z}_n^i; \theta^{i(t)}) \big(\mathbf{z}_n^i - \boldsymbol{\eta}_c^{i(t+1)}\big)\big(\mathbf{z}_n^i - \boldsymbol{\eta}_c^{i(t+1)}\big)^\top}{\sum_{n=1}^{N_i} p(c_n^i = c \mid \mathbf{z}_n^i; \theta^{i(t)})}. \tag{36}$$

In practice, we initialize the mixture weights as uniform ($\pi_c^i = 1/C$), set $\boldsymbol{\eta}_c^i$ using $k$-means centroids, and initialize $\Sigma_c^i$ as diagonal matrices. The EM algorithm is iterated until convergence of $\ell(\theta^i)$.

## F. Consistency between mixture of GRFNs and GMM

The consistency between Gaussian mixture models (GMM) and mixture Gaussian random fuzzy numbers (m-GRFN) can be clarified by their component structures. In a GMM, each observation is generated from one of $C$ Gaussian components,

$$p(\mathbf{z}) = \sum_{c=1}^{C} \pi_c \mathcal{N}(\mathbf{z}; \boldsymbol{\eta}_c, \Sigma_c), \tag{37}$$

where $\pi_c$ denotes the prior weight. Each component describes the feature distribution under a conditional probability. In the evidential framework, the same conditional design is retained, but each Gaussian is replaced with a Gaussian random fuzzy number (GRFN) that enriches the probabilistic description with evidential precision. Hence,

$$\tilde{Y} \sim \sum_{c=1}^{C} \pi_c \tilde{\mathcal{N}}(\mu_c, \sigma_c^2, h_c). \tag{38}$$

If the distributional features of each component can be consistently mapped to evidential quantities, then the applicability of m-GRFN naturally coincides with that of GMM. Both models share the same mixture structure and rely on prior weights for aggregation, ensuring aligned application scenarios.

To be noticed, when the evidential precision tends to infinity ($h_c \to +\infty$), the GRFN degenerates into a standard Gaussian random variable. In this limit, the entire m-GRFN reduces exactly to the original GMM. This shows that GMM can be viewed as a special case of m-GRFN, while m-GRFN itself provides a principled evidential generalization of the classical mixture modeling paradigm.

## G. Experimental details

### G.1. Datasets

We evaluate DPsurv on five cancer types from The Cancer Genome Atlas (TCGA), where $n$ denotes the number of patients and WSI denotes the number of whole-slide images: Breast Invasive Carcinoma (BRCA, $n = 1{,}041$, WSI=1,111), Bladder Urothelial Carcinoma (BLCA, $n = 373$, WSI=437), Uterine Corpus Endometrial Carcinoma (UCEC, $n = 504$, WSI=565), Kidney Renal Clear Cell Carcinoma (KIRC, $n = 511$, WSI=517), and Lung Adenocarcinoma (LUAD, $n = 456$, WSI=1,024). During preprocessing, we removed duplicate cases and excluded samples with missing survival time.

### G.2. Baselines

For baseline training, we use the AdamW optimizer with a learning rate of $1 \times 10^{-4}$, a weight decay of $1 \times 10^{-5}$, and a cosine learning-rate decay scheduler. Supervised baselines are trained with the negative log-likelihood loss for 20 epochs using a batch size of one patient. For PANTHER, we instead employ the Cox proportional hazards loss, training for 50 epochs with a batch size of 64 patients.

We compare DPsurv with the following representative methods:

- **ABMIL**: An attention-based MIL framework originally designed for WSI classification; in this work, we adapt it for discrete survival prediction.

- **TransMIL**: A Transformer-based MIL model for WSI classification; here, we modify it for discrete survival prediction.

- **DSMIL**: A dual-stream MIL model originally for WSI classification; we adapt it to discrete survival prediction.

- **ILRA**: An instance-level representation aggregation MIL model, developed for classification, and extended here to *discrete survival prediction*.

- **BayesMIL**: A Bayesian MIL framework proposed for WSI classification, adapted in our study for discrete survival prediction.

- **AttnMISL**: A survival-specific MIL model; for consistency with other methods, we implement it in the discrete survival prediction setting.

- **PANTHER**: A prototype-based survival model; we also adapt it to the discrete survival prediction setting for consistent evaluation.

- **EVREG**: An evidential regression model originally proposed for tabular data; in our setting, WSI features are reduced by PCA and clustered via GMM before applying EVREG with its original loss function.

- **UMSA**: A multiscale survival model guided by genomic data; since molecular inputs are unavailable in our setting, we adapt it to a single-scale version using only WSI features.

For all baselines, we unify the survival time discretization into a four-class quantile setting, consistent with traditional survival formulations and enabling direct probability outputs for calibration assessment.

### G.3. Evaluation Criteria

To comprehensively evaluate survival prediction models, we employ three widely used metrics: the concordance index (C-index), the integrated Brier score (IBS), and the integrated binomial log-likelihood (IBLL). Together, these metrics assess both the discriminative ability and calibration quality of survival estimates under censoring.

**Concordance Index.** The C-index quantifies a model's discriminative power by measuring the agreement between predicted risks and actual survival outcomes. It is defined as the proportion of all comparable subject pairs whose predictions are correctly ordered:

$$\text{C-index} = \frac{\sum_{i,j} \mathbf{1}(T_i < T_j) \, \mathbf{1}(\hat{r}_i > \hat{r}_j)}{\sum_{i,j} \mathbf{1}(T_i < T_j)}, \tag{39}$$

where $T_i$ and $T_j$ are observed times, $\hat{r}_i$ is the predicted risk score, and $\mathbf{1}(\cdot)$ is the indicator function. A C-index of $0.5$ indicates random ranking, while values closer to $1$ suggest near-perfect concordance.

**Integrated Brier Score.** The Brier score (BS) measures the squared error between predicted survival probabilities $\hat{S}(t|x_i)$ and observed binary survival outcomes at a fixed time $t$. To account for right censoring, inverse-probability weights are introduced via the Kaplan–Meier estimate $\hat{G}(\cdot)$ of the censoring distribution:

$$\text{BS}(t) = \frac{1}{N} \sum_{i=1}^{N} \left[ \frac{\hat{S}(t|x_i)^2 \, \mathbf{1}(T_i \leq t, D_i = 1)}{\hat{G}(T_i)} + \frac{(1 - \hat{S}(t|x_i))^2 \, \mathbf{1}(T_i > t)}{\hat{G}(t)} \right], \tag{40}$$

where $D_i$ is the event indicator ($D_i = 1$ if an event is observed for subject $i$ and $D_i = 0$ if censored), and $\hat{G}(\cdot)$ is the Kaplan–Meier estimate of the censoring distribution. Aggregating BS over a time interval $[t_1, t_2]$ yields the integrated Brier score:

$$\text{IBS} = \frac{1}{t_2 - t_1} \int_{t_1}^{t_2} \text{BS}(s) \, ds. \tag{41}$$

A lower IBS value indicates more accurate and better calibrated survival probability predictions.

**Integrated Binomial Log-Likelihood.** The binomial log-likelihood (BLL) evaluates the fit of predicted probabilities to observed outcomes and, analogously to the Brier score, incorporates inverse censoring weights via $\hat{G}(\cdot)$:

$$\text{BLL}(t) = \frac{1}{N} \sum_{i=1}^{N} \left[ \frac{\log\big(1 - \hat{S}(t|x_i)\big) \, \mathbf{1}(T_i \leq t, D_i = 1)}{\hat{G}(T_i)} + \frac{\log\big(\hat{S}(t|x_i)\big) \, \mathbf{1}(T_i > t)}{\hat{G}(t)} \right]. \tag{42}$$

The integrated version over $[t_1, t_2]$ is given by

$$\text{IBLL} = \frac{1}{t_2 - t_1} \int_{t_1}^{t_2} \text{BLL}(s) \, ds. \tag{43}$$

### G.4. DPsurv

**Evidential Neural Network Initialization.** We initialize the model by a weighted K-means algorithm. For each Gaussian component, we drop samples with small mixture proportions ($\hat{\pi}_c \leq \tau$), $\ell_2$-normalize the remaining features, and run weighted K-means with $\hat{\pi}_c$ as sample weights. The resulting cluster centroids serve as slide-level prototype vectors $\boldsymbol{p}_{c,k}$. For each slide-level prototype, we aggregate the survival responses of its assigned samples to initialize the evidential parameters: set $\beta_{c,k0}$ to the weighted mean of the log survival time, $\sigma_{c,k}^2$ to the corresponding weighted variance, and initialize $\beta_{c,k} = 0$. The epistemic uncertainty parameter $h_{c,k}$ is set to be 4 and scaled by the average mixture proportion within the cluster, thereby discounting the evidence of slide-level prototypes. The decay parameter $\gamma_{c,k}$ is initialized proportional to the inverse square root of the average squared cosine distance within the cluster. Note that with the default $C = 16$ patch prototypes, a given slide typically activates only a subset of them, since tissue components absent from that slide receive near-zero mixture weight $\hat{\pi}_c$ and are therefore omitted from visualization.

**Hyperparameter Settings.** DPsurv uses a unified learning rate of 1e-4. $K$ is selected from $\{1, 2, 3, 4\}$ by applying early stopping on the validation set. All baselines are retrained under the same validation/early-stopping protocol using their originally reported optimal configurations.

For model training, we set the number of patch prototypes to 16 and initialize $k$-means prototypes using 100,000 sampled patches. The AdamW optimizer with a cosine learning rate scheduler is adopted across all experiments. All experiments are conducted on NVIDIA A100 GPUs with 80GB memory. The detailed hyperparameter configurations for different datasets are summarized in Table 5.

*Table 5.* Hyperparameter settings for different datasets. All datasets share the same configuration except for the prototype-initialization threshold $\tau$. The number of component prototypes $K$ is selected per fold by early stopping and is therefore not listed here.

| Dataset | Learning rate | Epoch | Batch size | Weight decay | $\tau$ | $\lambda$ |
|---------|---------------|-------|------------|--------------|--------|-----------|
| BLCA | 1e−4 | 30 | 32 | 2e−4 | 0.01 | 0.5 |
| KIRC | 1e−4 | 30 | 32 | 2e−4 | 0.1 | 0.5 |
| UCEC | 1e−4 | 30 | 32 | 2e−4 | 0.1 | 0.5 |
| BRCA | 1e−4 | 30 | 32 | 2e−4 | 0.01 | 0.5 |
| LUAD | 1e−4 | 30 | 32 | 2e−4 | 0.01 | 0.5 |

**Prototype initialization threshold $\tau$.** The threshold $\tau$ is the only hyperparameter adapted to the characteristics of different cancer cohorts. It is used during prototype initialization to filter out samples whose assignment probability $\pi$ falls below $\tau$. In cancers with low morphological heterogeneity, assignment probabilities are generally higher and more concentrated, which allows the use of a larger $\tau$ (0.1) to discard low-confidence samples and improve the reliability of evidence associated with each prototype. Conversely, in highly heterogeneous cancers, where assignment probabilities are more dispersed, a smaller $\tau$ (0.01) is preferred to retain sufficient diversity in the initialization.

## H. Detailed Sensitivity Analysis

We conduct a hyperparameter sensitivity analysis to study the robustness of our proposal with those parameters on two cancer datasets, KIRC and UCEC. The results shown in Tables 6–7 indicate that performance varies slightly across a broad range of these settings in 5-fold cross-site validation, suggesting that DPsurv is robust and does not require heavy hyperparameter tuning. For example, the C-index for KIRC remains within a relatively tight band of approximately 0.71–0.73 across many choices of $K$, $C$, $\lambda$ and $\alpha$, and UCEC similarly stays around 0.71–0.74.

## I. Extended Interpretability Analysis

### I.1. Additional examples

To reduce concerns of cherry-picking, we include two additional randomly selected whole-slide examples in Figure 5 and Figure 6. Each example presents the assignment map and relative risk visualization.

### I.2. Expert evaluation

We additionally performed a small-scale, pilot clinical coherence evaluation. A subset of slides was randomly sampled, and for each case we generated both the GMM-based assignment map and the corresponding relative risk visualization produced by DPsurv. Two board-certified pathologists independently assessed the coherence between the model-derived interpretations and their own clinical judgement, using a 10-point ordinal scale.

As summarized in Table 8, the coherence scores are consistently high across cases. Furthermore, the minimal difference between the average ratings for the assignment map and the risk visualization indicates that when the GMM-derived component assignments align well with pathologists' expectations, the downstream risk estimation produced by DPsurv is also more likely to be clinically reliable. This provides additional evidence supporting the interpretability and clinical plausibility of our model's reasoning process.

*Table 6.* Hyperparameter sensitivity analysis on KIRC dataset. Metrics are reported as mean ± standard deviation.

| Parameter | Setting | C-index | IBS | IBLL |
|---|---|---|---|---|
| $K$ | 2 | 0.724±0.08 | 0.174±0.03 | 0.521±0.08 |
| | 3 | 0.725±0.07 | 0.187±0.02 | 0.546±0.05 |
| | 4 | 0.732±0.10 | 0.207±0.03 | 0.592±0.07 |
| | 5 | 0.712±0.09 | 0.229±0.03 | 0.646±0.07 |
| | 6 | 0.714±0.08 | 0.238±0.03 | 0.671±0.08 |
| $C$ | 8 | 0.725±0.07 | 0.237±0.04 | 0.665±0.09 |
| | 16 | 0.732±0.10 | 0.207±0.03 | 0.592±0.07 |
| | 32 | 0.719±0.08 | 0.192±0.03 | 0.557±0.07 |
| $\lambda$ | 0.1 | 0.733±0.08 | 0.440±0.07 | 1.314±0.22 |
| | 0.3 | 0.732±0.08 | 0.298±0.07 | 0.830±0.16 |
| | 0.5 | 0.732±0.10 | 0.207±0.03 | 0.592±0.07 |
| | 0.7 | 0.724±0.09 | 0.166±0.03 | 0.499±0.08 |
| | 0.9 | 0.715±0.09 | 0.162±0.06 | 0.503±0.16 |
| $\alpha$ | 0.1 | 0.735±0.09 | 0.186±0.02 | 0.545±0.05 |
| | 0.3 | 0.734±0.09 | 0.194±0.02 | 0.563±0.04 |
| | 0.5 | 0.732±0.10 | 0.207±0.03 | 0.592±0.07 |
| | 0.7 | 0.728±0.08 | 0.231±0.04 | 0.649±0.08 |
| | 0.9 | 0.715±0.09 | 0.305±0.04 | 0.833±0.10 |

*Table 7.* Hyperparameter sensitivity analysis on UCEC dataset. Metrics are reported as mean ± standard deviation.

| Parameter | Setting | C-index | IBS | IBLL |
|---|---|---|---|---|
| $K$ | 2 | 0.735±0.08 | 0.117±0.04 | 0.379±0.09 |
| | 3 | 0.728±0.11 | 0.147±0.04 | 0.455±0.09 |
| | 4 | 0.722±0.09 | 0.169±0.03 | 0.502±0.08 |
| | 5 | 0.715±0.08 | 0.186±0.03 | 0.537±0.08 |
| | 6 | 0.722±0.08 | 0.190±0.03 | 0.545±0.08 |
| $C$ | 8 | 0.740±0.08 | 0.159±0.03 | 0.480±0.08 |
| | 16 | 0.728±0.11 | 0.147±0.04 | 0.455±0.09 |
| | 32 | 0.722±0.11 | 0.144±0.04 | 0.447±0.10 |
| $\lambda$ | 0.1 | 0.742±0.10 | 0.399±0.10 | 1.113±0.31 |
| | 0.3 | 0.741±0.10 | 0.235±0.05 | 0.651±0.12 |
| | 0.5 | 0.728±0.11 | 0.147±0.04 | 0.455±0.09 |
| | 0.7 | 0.727±0.10 | 0.113±0.03 | 0.371±0.09 |
| | 0.9 | 0.721±0.10 | 0.110±0.04 | 0.364±0.12 |
| $\alpha$ | 0.1 | 0.722±0.09 | 0.120±0.03 | 0.390±0.09 |
| | 0.3 | 0.727±0.10 | 0.130±0.03 | 0.414±0.08 |
| | 0.5 | 0.728±0.11 | 0.147±0.04 | 0.455±0.09 |
| | 0.7 | 0.723±0.09 | 0.174±0.04 | 0.514±0.10 |
| | 0.9 | 0.709±0.09 | 0.255±0.05 | 0.701±0.12 |

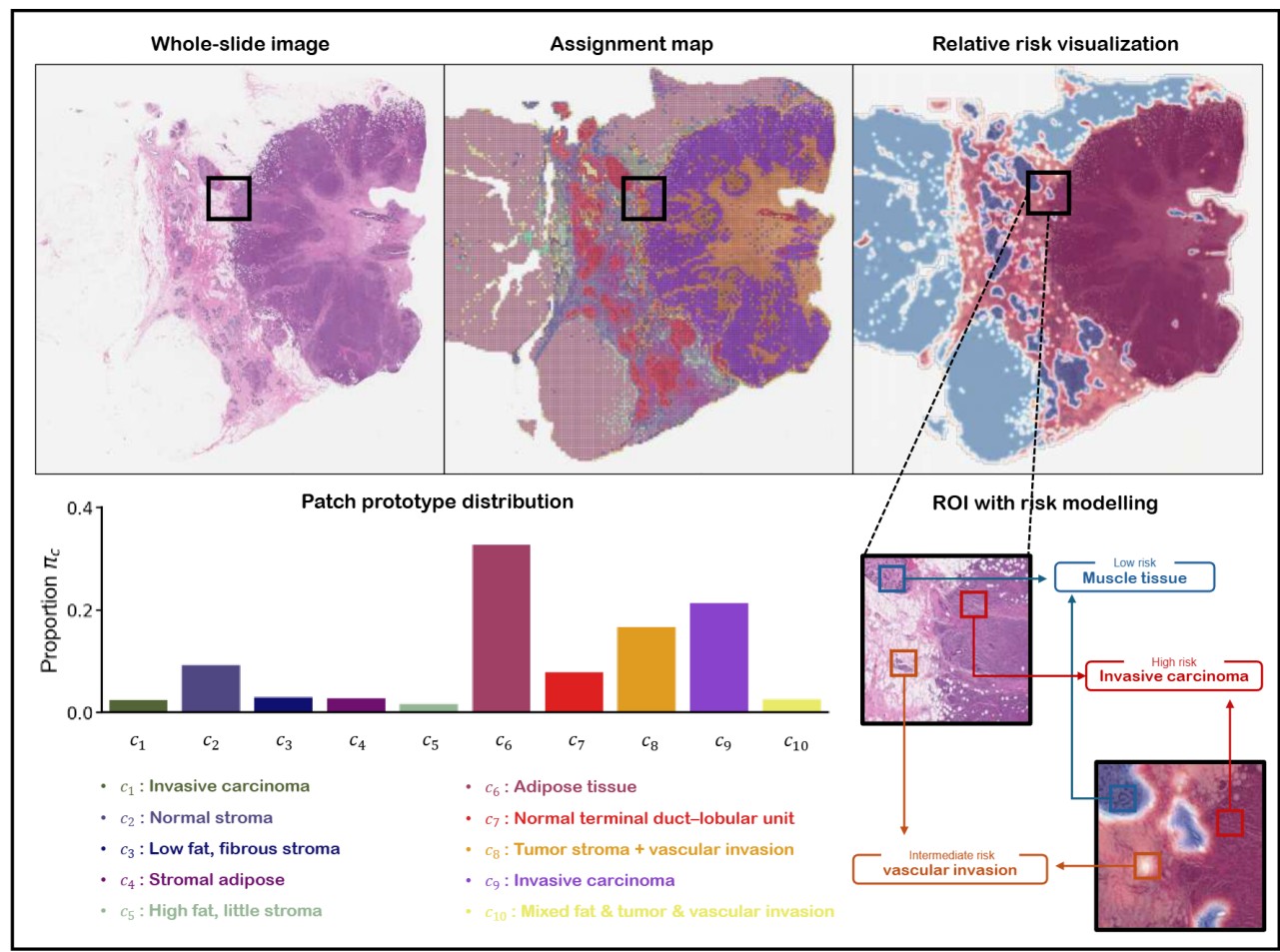

*Figure 5.* Additional interpretation of the DPsurv in WSI survival prediction.

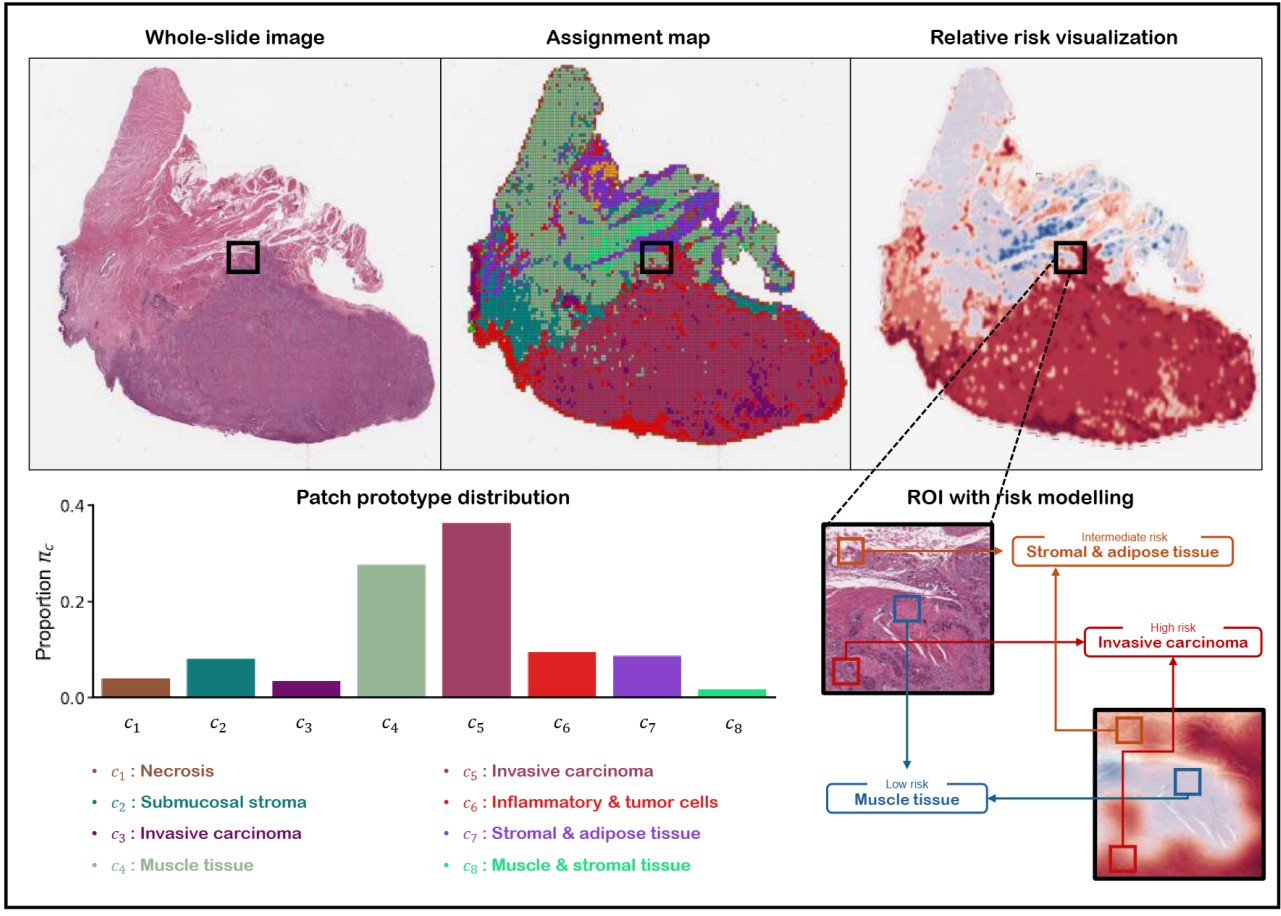

*Figure 6.* Additional interpretation of the DPsurv in WSI survival prediction.

*Table 8.* Coherence between model-generated interpretations and clinical insights.

| Case ID | Relative Risk Visualization | Assignment Map |
|---|---|---|
| TCGA-4Z-AA7S | 7 | 8 |
| TCGA-93-8067 | 9 | 9 |
| TCGA-A2-A3XW | 8 | 9 |
| TCGA-UY-A78L | 7 | 9 |
| TCGA-XF-A8HB | 7 | 8 |
| TCGA-XF-A9SH | 7 | 8 |
| TCGA-XF-A9ST | 8 | 9 |
| TCGA-XF-AAMJ | 4 | 8 |
| TCGA-XF-AAMZ | 7 | 7 |
| TCGA-ZF-AA4W | 6 | 7 |
| TCGA-05-4382 | 7 | 7 |
| TCGA-05-4402 | 8 | 8 |
| TCGA-55-7903 | 7 | 9 |
| TCGA-69-8255 | 8 | 8 |
| TCGA-75-7027 | 7 | 7 |
| TCGA-75-7031 | 7 | 7 |
| TCGA-86-8358 | 7 | 9 |
| **Overall Average** | **7.1** | **8.1** |

## J. Toy example

Consider two component prototypes associated with component $c$,

$$\tilde{Y}_{c,1} \sim \tilde{N}(\mu_{c,1}, \sigma_{c,1}^2, h_{c,1}), \qquad \tilde{Y}_{c,2} \sim \tilde{N}(\mu_{c,2}, \sigma_{c,2}^2, h_{c,2}),$$

together with their similarity scores $s_{c,1}, s_{c,2} \in [0, 1]$. Using the fusion rule, the fused GRFN is

$$\tilde{Y}_c \sim \tilde{N}(\mu_c, \sigma_c^2, h_c),$$

with parameters

$$h_c = s_{c,1} h_{c,1} + s_{c,2} h_{c,2}, \tag{44}$$

$$\mu_c = \frac{s_{c,1} h_{c,1} \mu_{c,1} + s_{c,2} h_{c,2} \mu_{c,2}}{s_{c,1} h_{c,1} + s_{c,2} h_{c,2}}, \tag{45}$$

$$\sigma_c^2 = \frac{s_{c,1}^2 h_{c,1}^2 \sigma_{c,1}^2 + s_{c,2}^2 h_{c,2}^2 \sigma_{c,2}^2}{(s_{c,1} h_{c,1} + s_{c,2} h_{c,2})^2}. \tag{46}$$

For example, Let

$$\mu_{c,1} = 2, \ \mu_{c,2} = 7, \qquad \sigma_{c,1}^2 = 1, \ \sigma_{c,2}^2 = 4,$$
$$h_{c,1} = 1.2, \ h_{c,2} = 0.8, \qquad s_{c,1} = 0.6, \ s_{c,2} = 0.3.$$

Substituting into (44)–(46) yields

$$h_c = 0.96, \qquad \mu_c = 3.25, \qquad \sigma_c^2 \approx 0.812.$$

Thus, the fused GRFN is

$$\tilde{Y}_c = \tilde{N}(3.25, \ 0.812, \ 0.96).$$

## K. Comparison with alternative uncertainty quantification approach

We conduct extra experimental comparisons of our method (without early stopping of $K$) and two strong non-evidential uncertainty baselines, Monte-Carlo (MC) dropout and Deep ensemble. The comparison results in Table 9 show superior predictive performance across all three evaluation metrics compared with the MC dropout and Deep ensemble. To make it clearer, we highlight the key differences between GRFN-based uncertainty modeling and traditional non-evidential baselines here:

(1) **Disentangling epistemic and aleatoric uncertainty.** MC Dropout and Deep Ensemble estimate uncertainty via parameter sampling or model ensembling, but they do not distinguish between epistemic uncertainty and aleatoric uncertainty. In contrast, GRFN $\tilde{Y} \sim \tilde{\mathcal{N}}(\mu, \sigma^2, h)$ explicitly models both uncertainties at the evidence level, i.e., $\sigma$ and $h \in [0, +\infty)$ represent the aleatoric and epistemic uncertainties.

(2) **Predictive calibration through uncertainty weighting.** While MC Dropout and Deep Ensemble only quantify uncertainty, GRFN not only estimates uncertainty but also leverages it to optimize model behavior. Specifically, evidences with higher uncertainty are automatically assigned lower weights during aggregation, enabling GRFN to make more robust and interpretable predictions.

*Table 9.* Comparison of uncertainty modeling approaches on survival prediction.

| Method | C-index | IBS | IBLL |
|---|---|---|---|
| MC Dropout | 0.645 | 0.774 | 2.820 |
| Deep Ensemble | 0.648 | 0.803 | 3.505 |
| **Ours** | **0.687** | **0.209** | **0.604** |

In addition, we provide calibration plots comparing DPsurv with two widely used uncertainty estimation baselines, MC Dropout and Deep Ensemble. As shown in Fig. 7, we report the proportion of $\alpha$-level BPIs and PPIs that contain the uncensored survival times for $\alpha \in 0.1, \ldots, 0.9$ across the five TCGA cohorts (BRCA, BLCA, LUAD, UCEC, and KIRC). These results further demonstrate that DPsurv achieves consistently improved calibration performance relative to both baselines.

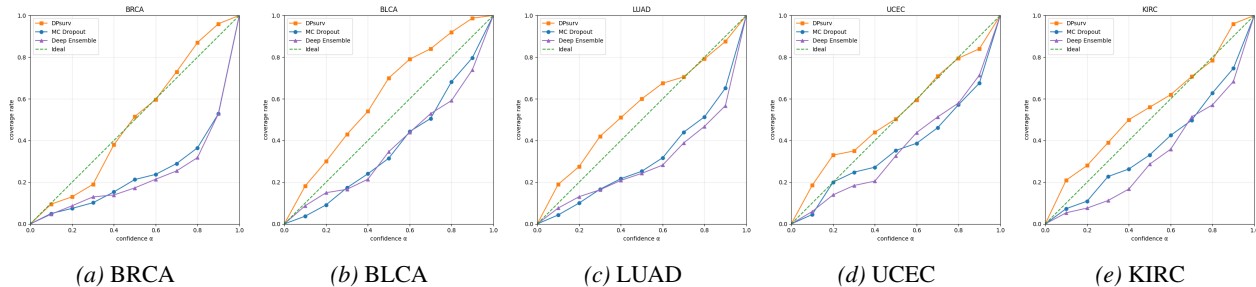

*(a)* BRCA     *(b)* BLCA     *(c)* LUAD     *(d)* UCEC     *(e)* KIRC

*Figure 7.* Calibration plots comparing DPsurv with MC Dropout and Deep Ensemble across five TCGA datasets. The plots show the empirical coverage of $\alpha$-level BPIs and PPIs with respect to uncensored survival times for $\alpha \in 0.1, \ldots, 0.9$. Curves closer to the diagonal indicate better calibration performance.

## L. Limitations

While DPsurv achieves strong discriminative and calibration performance, several limitations remain. First, as a targeted failure analysis, we examined TCGA-XF-AAMJ, the lowest-scoring case in our clinical coherence study (Table 8): when the tumor component occupies only a small proportion of the slide, its risk evidence is heavily discounted, increasing epistemic uncertainty and challenging reliable risk ranking, indicating that slides dominated by non-tumor tissue may yield less stable estimates. Second, since our experiments adopt site-stratified cross-validation, the results support robustness under moderate site-level variation rather than formal out-of-distribution detection, leaving the behavior under stronger domain shifts and the potential bias in component-level interpretations to be studied. Finally, our clinical interpretability assessment is a pilot-scale study intended to validate the semantic plausibility of the learned prototypes rather than to provide definitive clinical validation, and larger prospective studies are needed before deployment.

## M. Discussion

Our proposed DPsurv framework not only provides accurate survival prediction but also offers interpretability through prototype-based evidence modeling. By identifying representative patch and component prototypes, the model can uncover potential histological phenotypes that correspond to typical morphological patterns. Such discoveries may aid pathologists in establishing new classification rules for differentiating subtypes within the same tissue type, thereby facilitating more refined diagnostic systems. Furthermore, because our model quantifies the risk associated with each morphological phenotype, it has the potential to reveal hidden prognostic factors. This ability to link specific tissue patterns with survival risk highlights the clinical utility of DPsurv, suggesting that it may serve as a valuable tool for both precision prognosis and exploratory pathology research.

