# OpenReview forum: "DPsurv: Dual-Prototype Evidential Fusion for Uncertainty-Aware and Interpretable Whole Slide Image Survival Prediction"
_ICML.cc/2026/Conference — ICML 2026 regular_

### Official Review · Reviewer_8mzg · 2026-02-26

**Soundness:** 3
**Presentation:** 2
**Significance:** 2
**Originality:** 2
**Overall Recommendation:** 4
**Confidence:** 4

**Summary:**

This paper proposes DPsurv, a dual-prototype evidential fusion network for whole-slide image (WSI) survival prediction. The core innovation lies in employing a patch prototype-guided Gaussian Mixture Model (GMM) for component-level slide representation, combined with Gaussian Random Fuzzy Numbers (GRFNs) for evidence-based uncertainty modeling. The framework outputs interval-valued survival predictions that explicitly disentangle aleatoric and epistemic uncertainty, while achieving multi-level interpretability through patch prototype assignment maps, component prototype evidence reasoning, and component-wise relative risk visualization. Experiments on five TCGA cancer cohorts demonstrate state-of-the-art performance in both discriminative accuracy and calibration, and the clinical coherence of the interpretations has been validated by board-certified pathologists.

**Compliance With Llm Reviewing Policy:**

Affirmed.

**Final Justification:**

I thank the authors for their thoughtful rebuttal. The additional clarification addresses part of my earlier concerns and makes the paper’s intended contribution clearer.

I still retain some reservations, especially regarding the overall novelty and the strength of the claims. However, I acknowledge that the rebuttal substantially improved the explanation of the method and reduced some of the ambiguity that affected my initial assessment.

Accordingly, I raise my score by one point, although I remain somewhat cautious in my overall evaluation.

**Key Questions For Authors:**

1. Appendix D states that the GMM parameters are fixed and do not participate in training of the subsequent evidential fusion network, yet Section 3.1 describes DPsurv as an "end-to-end survival-evidence pipeline." Could the authors clarify which parameters are jointly optimized and which are fixed at each stage? A concise description of the full training procedure in the main text would improve methodological clarity.
2. In Table 1, adding CP alone (without EM) causes the C-index to drop substantially from 0.663 to 0.538. Could the authors provide a more detailed explanation for this behavior? Does this degradation occur consistently across all five individual datasets, and what does this imply about the dependency between the CP and EM modules?
3. Regarding the fold-varying K setting (e.g., K=1–4 for BLCA): could the authors describe the specific procedure by which K is chosen for each fold? In particular, is K determined purely based on prior knowledge of the dataset characteristics, or is any form of validation performance consulted during this selection?
4. The parameter λ is set to 0.5 uniformly across all datasets. However, Table 4 shows that λ=0.7 yields better IBS/IBLL on KIRC while λ=0.5 achieves better C-index. In a clinical deployment scenario where both discrimination and calibration matter, what criterion would you recommend for practitioners to select λ?

**Limitations:**

1. The staged training procedure (fixed GMM followed by evidential fusion training) sits in tension with the "end-to-end" description in the main text. Clarifying this distinction explicitly in the main text rather than deferring to the appendix would improve methodological transparency.

2. All experiments rely on UNI2-h as the sole feature extractor. Given that GMM fitting quality is sensitive to the geometric structure of the feature space, a brief discussion of how results might vary under other commonly used extractors (e.g., CONCH, PLIP) would help assess the generalizability of the approach.

3. The paper assumes that patch embeddings follow a Gaussian mixture distribution (Equation 3). While this is a standard and widely used assumption, a brief discussion of its appropriateness for WSI patch embeddings across the five cancer types examined — potentially supported by a visualization of the feature distributions — would strengthen the theoretical grounding of the method.

**Strengths And Weaknesses:**

Strengths:
This paper simultaneously addresses two important yet underexplored challenges in computational pathology-interpretability and uncertainty quantification in survival analysis. Incorporating GRFNs into a WSI survival analysis framework is technically rigorous and novel, enabling principled disentanglement of epistemic and aleatoric uncertainty, a capability absent in all compared baselines. The substantial improvement in calibration metrics (IBS/IBLL) over all baselines, including other uncertainty-aware methods such as BayesMIL and EVREG, is a particularly noteworthy result that highlights the practical advantage of the evidential framework. The three-level interpretability framework (feature, reasoning, and decision levels) is well-motivated, and the pathologist validation provides meaningful clinical grounding. The empirical evaluation is thorough, demonstrating consistent improvements in C-index, IBS, and IBLL over strong baselines across five diverse public cancer datasets.

Weaknesses:
1. The C-index drop caused by the CP module in Table 1 requires further explanation. The ablation results show that adding Component Prototypes (CP) without Evidence Mixture (EM) causes the C-index to drop from 0.663 to 0.538, approaching random-chance level. The paper attributes this to CP and EM needing to work jointly, which is a reasonable intuition, but no concrete mechanistic explanation is provided. It would strengthen the paper to clarify whether this degradation occurs consistently across all five datasets or primarily in certain cancer types, and to provide a brief theoretical account of why CP alone is insufficient for discrimination without the EM aggregation mechanism.
2. The K selection procedure and sensitivity analysis coverage could be made more transparent. Table 3 sets K as a fold-varying range (e.g., BLCA: K=1–4, LUAD: K=2–3), justified by training fold size considerations. While this is a reasonable engineering choice, the paper would benefit from a clearer description of exactly how K is determined in practice — specifically, whether K is selected based solely on dataset-level prior knowledge or whether any validation set performance is consulted. Additionally, the sensitivity analysis in Appendix G covers only KIRC and UCEC. Extending the analysis to BRCA, the most heterogeneous and worst-performing dataset, would further strengthen the claim that DPsurv is robust to hyperparameter choices across diverse cancer types.
3. The scale and diversity of the pathological evaluation could be expanded. The clinical coherence evaluation in Table 7 covers 10 cases, with the majority sharing the TCGA-XF institutional prefix, suggesting relatively concentrated institutional provenance. While this is a reasonable pilot evaluation, 10 cases from a limited institutional range may not fully demonstrate generalizability. Additionally, the relative risk visualization score for case TCGA-XF-AAMJ is 4 out of 10, notably lower than its assignment map score of 8. A brief targeted discussion of such outlier cases would help readers better understand the conditions under which the model's risk visualization may be less reliable.

---

> ### Author Rebuttal · Authors · 2026-03-30
>
> We thank the reviewer for the constructive feedback and recognition of the paper's strengths in calibration, uncertainty-aware reasoning, and multi-level interpretability. We address each concern below.
>
> > Response to W1: CP/EM interaction in the ablation
>
> Thanks for your detailed observation. We have added in the ablation study that CP and EM are not independently beneficial.
> CP introduces component-level evidence, but without EM this evidence is aggregated uniformly rather than by $\pi_c$, allowing absent/negligible components to distort predictions. We provide the full mechanism analysis in our response to Reviewer mLLo (W1), and the **per-dataset ablation table** here.
>
> | Dataset | C-index | IBS | IBLL |
> |---------|---------|-----|------|
> | BLCA | 0.5698±0.0818 | 0.3065±0.0370 | 0.8429±0.0954 |
> | BRCA | 0.5665±0.0925 | 0.2668±0.0594 | 0.7540±0.1552 |
> | LUAD | 0.5386±0.0657 | 0.3885±0.1022 | 1.4852±0.6292 |
> | UCEC | 0.5110±0.0845 | 0.2526±0.0689 | 0.7602±0.1700 |
> | KIRC | 0.5041±0.1180 | 0.2872±0.0200 | 0.9223±0.1842 |
>
> The consistent degradation across all five cohorts confirms that the CP-without-EM failure is a structural aggregation mismatch, not dataset-specific noise. We have also included a brief theoretical overview of the dependency between CP and EM.
>
> > Response to W2: Transparency of K-selection
>
> Thanks for your suggestion. **We have included a validation set for K selection:** within each fold, 15% of training data is held out, K is selected from {1,2,3,4} by applying early stopping on validation set. All baselines are retrained under the same protocol. Detailed protocol description is in our response to Reviewer mLLo (W4) and full updated per-dataset results in Reviewer CuW2 (W1). We will describe this protocol in the main text rather than leaving K to appear as a dataset-level design choice.
>
> For the sensitivity analysis, under the revised experimental protocol, **BRCA achieves state-of-the-art performance and is the most stable (std 0.03).** Due to time constraints, we will include the sensitivity analysis for all five datasets in the revision.
>
> > Response to W3: Scope of clinical interpretation study
>
> The current clinical interpretation study is indeed a pilot evaluation. We have revised Table 7 as a small-scale coherence evaluation. Due to time constraints, we are not able to further expand this analysis at this stage. In the revision, we will involve pathologists to assess a larger and more diverse set of samples, as the scale and diversity of the pathological evaluation could be further expanded.
>
> For Case TCGA-XF-AAMJ, please refer to our response to the limitation raised by Reviewer mLLo. We have included this case as part of a targeted failure analysis.
>
> > Response to KQ1: Staged training vs. "end-to-end"
>
> DPsurv is not optimized end-to-end: the unsupervised GMM is fitted first and then fixed, after which the evidential fusion network is trained. What we meant by “end-to-end” is the interpretability chain from feature decomposition to evidence reasoning to final decision. We will revise Sec. 3.1 and clarify the two-stage training procedure in the main text.
>
> > Response to KQ2: CP without EM degradation
>
> Addressed in W1.
>
> > Response to KQ3: How is K selected?
>
> Addressed in W2.
>
> > Response to KQ4: How should $\lambda$ be selected?
>
> Thanks for your question. An advantage of DPsurv is that $\lambda$ controls decision-risk preference. Larger $\lambda$ yields more conservative predictions by emphasizing the belief bound, while smaller $\lambda$ places more weight on plausibility. We use $\lambda=0.5$ as a neutral default with no prior preference. In the revision, we will clarify that larger $\lambda$ may be preferred for risk-averse settings such as clinical screening, whereas $\lambda=0.5$ is more appropriate when discrimination is the primary goal.
>
> > Response to Limitations
>
> We thank the reviewer for pointing out the limitation discussion.
>
> 1. We will clarify the “end-to-end” description in the main text.
>
> 2. We additionally conduct experiment below on UCEC using CONCH [1], a state-of-the-art vision–language pathology foundation model, to assess feature extractor robustness.
>
> | Method | C-index | IBS | NBLL |
> |---|---|---|---|
> | ABMIL | 0.7565±0.045 | 0.6948±0.149 | 2.2436±0.743 |
> | TransMIL | 0.7494±0.062 | 0.7994±0.161 | 2.8079±0.871 |
> | PANTHER | 0.7319±0.113 | 0.7914±0.153 | 2.6246±0.867 |
> | BayesMIL | 0.7588±0.062 | 0.6658±0.138 | 1.9255±0.522 |
> | **DPsurv** | **0.7795±0.096** | **0.2387±0.119** | **0.6734±0.309** |
>
> DPsurv maintains the best performance under a different feature extractor, strengthening evidence of backbone robustness.
>
> 3. We will add a brief discussion of the appropriateness of using Gaussian mixture distribution for WSI patch embedding
>
> [1] Lu, M.Y., et al. "A visual-language foundation model for computational pathology." Nature Medicine 30.3 (2024): 863-874.

---

> > ### Author Rebuttal · Reviewer_8mzg · 2026-04-01
> >
> > Thank you for your further explanation.

---

> > > ### Author Response · Authors · 2026-04-02
> > >
> > > Thank you for acknowledging that the concerns have been fully resolved. We appreciate your constructive feedback throughout the review process and would be happy to address any remaining suggestions in the final version. As noted in option (a), we would be grateful if you could consider updating your score accordingly.

---

### Official Review · Reviewer_JyWq · 2026-03-06

**Soundness:** 2
**Presentation:** 1
**Significance:** 2
**Originality:** 3
**Overall Recommendation:** 4
**Confidence:** 3

**Summary:**

The authors present an approach, DPSurv, for survival prediction from WSIs that incorporates multiple levels of modeling with uncertainty quantification. Experiments are performed on 5 subsets of TCGA, achieving competitive performance with interpretability visualizations that highlight differing histological features.

**Compliance With Llm Reviewing Policy:**

Affirmed.

**Final Justification:**

The authors have addressed my major concern regarding cross validation rigor and I have updated my score accordingly. Concerns around presentation, clarity, and method complexity are challenging to fully evaluate without an updated manuscript version.

**Key Questions For Authors:**

Please see weaknesses.

**Limitations:**

Limitations are not discussed.

**Strengths And Weaknesses:**

Strengths
1. The paper targets an important problem - survival prediction from WSIs while offering interpretability insights and uncertainty quantification.
2. Selection of datasets and baselines are reasonable.
3. Some ablation and sensitivity analyses are presented.

Weaknesses
1. I found the Methods and overall approach very difficult to follow. There are many different terms that are used, which are not not precisely defined and often overlap/clash across sections (e.g. prototype vs component vs component prototype; component evidence modeling vs component evidence mixture; etc).
2. Some sections in the Results are also hard to follow and the results used to support some claims are unclear. For instance, the sensitivity analysis section states that "DPsurv shows stable performance.." but does not refer to any figures or results until the last sentence which states that "Further sensitivity analyses and discussion are provided in Appendix G."
3. Critically, it appears that different sets of hyperparameters were needed for each of the 5 datasets with DPSurv (Appendix F.4). These hyperparameter variations not only include parameters specific to DPSurv, but also basic parameters like learning rate. However, all of the baselines were trained with the same learning rate. This creates a strong risk of performance bias and results inflation, especially given the marginal improvement of DPSurv over the baselines as is.

---

> ### Author Rebuttal · Authors · 2026-03-30
>
> We thank the reviewer for highlighting the presentation and transparency issues. We address concerns on terminology, organization, model selection, and limitations.
>
> > Response to W1: Terminology clarity
>
> This point relates to presentation clarity.  **We provid a terminology table** below to distinguish clearly among patch prototypes, GMM components, component prototypes, component evidence, and component evidence mixture. We will add this table at the beginning of Section 3.
>
> | Term | Definition | Use / Role in DPsurv | Where used |
> |------|-----------|----------------------|------------|
> | **Patch prototype** ($\mathbf{h}_c$) | Global centroid in patch embedding space, learned via K-means. Shared across all WSIs. There are C of them. | Used to partition patch embeddings into coarse morphological groups, such as tumor, stroma, or necrosis-like tissue patterns. They provide the basis for WSI decomposition. | Sec. 3.3 (Eq. 2–5) |
> | **Component** | A Gaussian mixture component aligned to a patch prototype. Each WSI has C components with parameters $(\hat{\pi}_c, \hat{\mu}_c, \hat{\Sigma}_c)$. | Represents one morphology-aware tissue component in the WSI, modeled as a Gaussian distribution in feature space. It summarizes the prevalence, center, and variation of one tissue pattern. | Sec. 3.3 (Eq. 3–4) |
> | **Component prototype** ($p_{c,k}$) | A learned vector in the slide embedding space. Each component c has K prototypes that serve as local risk experts. | Used to model finer-grained subtypes or risk patterns within one tissue component. They capture how different variants of the same tissue component relate to survival evidence. | Sec. 3.4 (Eq. 6–8) |
> | **Component evidence** | The GRFN Y_c produced by aggregating evidence from K component prototypes for component c. | Represents the survival risk evidence associated with one tissue component, including both predicted survival tendency and its uncertainty. | Sec. 3.4 (Eq. 7) |
> | **Component evidence mixture** | The m-GRFN that aggregates all C component-level GRFNs into a slide-level prediction. | Mixtures the risk evidence from all tissue components into a final slide-level survival representation and prediction. | Sec. 3.5 (Eq. 10) |
>
> In addition, we included a paragraph with the roadmap at the beginning of Sec. 3.1 to clarify the hierarchy of objects before moving to the technical details. More concretely, we clarify that DPsurv uses two prototype levels: globally shared patch prototypes first decompose a WSI into C morphology-aware GMM components, and then K component prototypes within each component act as local risk experts to generate component-level evidence, which is finally aggregated into a slide-level evidential prediction.
>
> > Response to W2: Organization of the Results section
>
> We recognize that the current Results section can be more readable. Specifically, in Sec. 4.2’s sensitivity analysis part, conclusions are presented before they are related to corresponding results. Currently, four parts are included in the Results section: discriminative performance, module contribution and robustness (ablation and sensitivity analysis), interpretability analysis, and calibration analysis.
>
> As for how we improve the Results section in the revision, we rearrange the structure of the Results section in a more logical way: main performance comparison, ablation analysis of module contributions, sensitivity analysis of robustness, interpretability analysis, and uncertainty analysis. Within each part, we make sure each conclusion is directly related to corresponding results in each figure/table and a takeaway is presented at the beginning of each part.
>
> > Response to W3: Fairness of hyperparameter selection
>
> Some hyperparameters may vary across settings due to differences in cancer types (i.e., heterogeneity) and training sample sizes in order to better capture potential component subtypes. **We have included a validation set for hyperparameter selection and re-run the main experiment.** The details of the experiment can be found in our response to Reviewer mLLo (W4) and the per-dataset results in our response to Reviewer CuW2 (W1). In this revised experiment, the best overall method in terms of mean C-index, IBS, and IBLL remains to be DPsurv. In the revised version, we have included the selection protocol in the main text so that hyperparameters are not presented as a design choice without a clear selection rule.
>
> > Response to Limitations discussion
>
> We have added a dedicated Limitations section clarifying that the current study supports robustness under moderate site-level variation rather than formal OOD detection, depends on the feature extractor and prototype granularity choices, and includes a pilot-scale clinical interpretation study rather than definitive clinical validation.

---

> > ### Author Rebuttal · Reviewer_JyWq · 2026-04-01
> >
> > I thank the authors for their response. Could the authors please clarify how the updated performance/hyperparameter selection was performed? Which hyperparameters were shared across all models, which ones were different across models and/or datasets, and which ones were chosen via hyperparameter selection?
> >
> > In comparing the performance reported here to that in reported in the PANTHER publication, I also see some differences. For instance, the original PANTHER publication reported a C-index of 0.758 for BRCA and 0.685 for LUAD which appear higher than the values reported for PANTHER (and DPSurv) in the current work. Could the authors clarify what may cause this discrepancy?

---

> > > ### Author Response · Authors · 2026-04-02
> > >
> > > We thank the reviewer for the continued engagement. **We are glad that we have addressed your concerns regarding the unclear presentation (W1, W2).** Regarding the fairness of comparison (W3), we thank you for your suggestions and we have included a validation set for hyperparameter selection and re-ran the main experiments in our first rebuttal round. The detailed experimental setup and updated results have been provided in our responses to Reviewer CuW2 (W1) and Reviewer mLLo (W4). We will also release full codebase including DPsurv and all baseline training scripts to facilitate reproducibility. Below we provide further clarification on your two specific follow-up questions.
> > >
> > > > Q1: Could the authors please clarify how the updated performance/hyperparameter selection was performed?
> > >
> > > **1. Which hyperparameters were shared across all models:**
> > >
> > > All models use the same feature extractor (UNI2-h), the same 5-fold site-stratified cross-validation splits, and the same early-stopping mechanism with a held-out validation set (15% of training data per fold).
> > >
> > > **2. Which ones were different across models and/or datasets:**
> > >
> > > For compared baseline methods, we follow the optimal parameter configuration reported in their original paper. For example, PANTHER uses Cox loss with batch size 64 and MIL baselines use NLL loss with batch size 1. All baseline hyperparameters are fixed across datasets.
> > >
> > > For DPsurv, all hyperparameters are fixed across datasets (learning rate = 1e-4, weight decay = 2e-4, α = 0.5, λ = 0.5, τ = 0.01, batch size = 32), with the exception of K (number of component prototypes), which varies across folds via hyperparameter selection.
> > >
> > > **3. Which ones were chosen via hyperparameter selection:**
> > >
> > > Only prototype number K in DPsurv is selected via validation, from the candidate set {1, 2, 3, 4} for each fold. K needs to adapt because the effective training size and subtype composition vary across folds. No other hyperparameter is tuned per dataset or per fold for any model.
> > >
> > > **4. On fairness of selecting K via validation for DPsurv but not for baselines:**
> > >
> > > 1. Using author-reported configurations for baselines is standard practice in computational pathology [1], as these configurations have been extensively tuned by the original authors and demonstrated stable performance across diverse datasets.
> > >
> > > 2. The unified early-stopping further ensures fair comparison.
> > >
> > > 3. We will release the full codebase including baseline training scripts to facilitate reproducibility.
> > >
> > > > Q2: Could the authors clarify what may cause this discrepancy?
> > >
> > > The discrepancy can be attributed to the following factors:
> > >
> > > 1. **Feature extractor.** We use UNI2-h (1536-dim) for all methods, whereas the original PANTHER paper uses UNI v1 (1024-dim). Differences in feature dimensionality and representation quality propagate to downstream prototype construction and survival performance.
> > >
> > > 2. **Data splits.** In our additional experiments, we hold out 15% of training data as a validation set for early stopping, which reduces the effective training size per fold compared to the original PANTHER setup.
> > >
> > > 3. **Early stopping.** All methods are evaluated at the point of best validation performance, which favors models with stronger generalizability. Under this protocol, PANTHER and BayesMIL emerge as the top-2 baselines, demonstrating their robustness.
> > >
> > > **This discrepancy is not specific to our work.** ProtoSurv [2] reproduced PANTHER with UNI features and different data splits, reporting C-index of 0.699 on BRCA and 0.631 on LUAD. The performance is lower than the original publication (0.758 and 0.685), confirming that differences in data splits can lead to notable performance variations. Importantly, all methods in our experiments are compared under the same unified protocol, ensuring a fair relative comparison.
> > >
> > > We thank you again for your comments and would be happy to address any remaining concerns in the final version.
> > >
> > > [1] Ramanathan, Vishwesh, et al. "Ensemble of prior-guided expert graph models for survival prediction in digital pathology." International Conference on Medical Image Computing and Computer-Assisted Intervention. Cham: Springer Nature Switzerland, 2024.
> > >
> > > [2] Wu, Junxian, et al. "Leveraging tumor heterogeneity: Heterogeneous graph representation learning for cancer survival prediction in whole slide images." Advances in Neural Information Processing Systems 37 (2024): 64312-64337.

---

### Official Review · Reviewer_mcnn · 2026-03-08

**Soundness:** 3
**Presentation:** 2
**Significance:** 2
**Originality:** 2
**Overall Recommendation:** 4
**Confidence:** 3

**Summary:**

The paper proposes a model that predicts survival from whole slide images. GRFN (Gaussian random fuzzy numbers) are used to model survival. The model uses some prototype patches shared across all slides in the dataset, enabling mechanistic interpretability.

The emprical results are convincing, but I initially rate this paper as 'weak reject' and I'm open to increase my score if my comments/concerns are addressed.

**Compliance With Llm Reviewing Policy:**

Affirmed.

**Final Justification:**

The authors' reply addressed the 2 major concerns that I had (i) a notational confusion and (ii) the differences to a parallel work EsurvFusion.

**Key Questions For Authors:**

I have several questions in the technical side
I recommend a major change in the way the method is presented, so it comprehensively addresses the following questions.

- In Eq. 2, a whole slide image is encoded by (i) measuring the similarity of each patch embedding to different prototypes and (ii) computing the average and (iii) concatenating them. Shouldn't this calculation involve a weighted sum of the prototype embeddings? weighted by the calculated similarities?
- Having discussed Eq. 2, in Eq. 4 there is a GRFN per component (or per prototype), which are simply concatenated and taken as the encoding the WSI. How does that related to Eq. 2?
- The prototypes $h_c$ are firstly indexed by {1,...,C} above Eq. 3, but in Sec. 3.4. they are indexed by {1,...,K}. In sum, the relationship between the GMM components and the prototypes is quite unclear to me.
- Below Eq. 7, we have that $h_c = \sum_{k=1}^K s_{c,k} h_{c,k}$, but there is $h_c$ both in the left hand side and right hand side. Does it mean there is a loop or fixed point equation in the pipeline?


My other question is about the specific novelty and distinctions of DPsurv.
For example EsurvFusion [1] uses GRFNs for survival prediction and in multi-modal settings, including histopathology images. How does DPsurv differe from [1]?

[1] Huang, Ling, et al. "EsurvFusion: An evidential multimodal survival fusion model based on Gaussian random fuzzy numbers." arXiv preprint arXiv:2412.01215 (2024).

**Limitations:**

Besides my questions/concerns that I mentioned above, I found many of the claims in the introduction/related works inaccurate.
For example:
- "... Moreover, uncertainty research has been primarily focused on classification models for discrete output ...": Most of the mentioned methods (Monte-carlo dropout, ensembling, Bayesian experimental design, etc.) are already usable for, e.g., regression.
- "... Although these methods address the gigapixel scale, tissue heterogeneity remains underexplored.": This is also not correct. For example attention-based models can discard big tissue regions and only focus on subtle tissue regions.

**Strengths And Weaknesses:**

Strenghts:
- As highlighted in the paper, GRFNs are a good choice for survival prediction (due to, e.g., providing beliefs and plausibility in Eq. 6)
- The method works with some prototype patches, shared across the dataset instances, thereby enabling interpretability.
- The empirical results are convincing. For example the experiments of Table 2 show the contribution of each model component to the improvement in performance.

Weaknesses:
- In the technical side and the presentation of the method, I have several questions that I've provided below in  "Key Questions For Authors".

---

> ### Author Rebuttal · Authors · 2026-03-30
>
> We thank the reviewer for the careful technical reading and openness to revise the score. We appreciate the recognition that GRFNs are well motivated and empirical results are promising. The main issues raised concern the presentation of Eqs. 2/4/7, notation consistency, and the distinction from EsurvFusion. We address each below.
>
> > KQ1: Should Eq. 2 involve a weighted sum of prototype embeddings?
>
> Thanks for the question. We clarify that the similarity are used for **soft assignment**, not for feature pooling [1] . Specifically, they are normalized to obtain component responsibilities, which are then used to estimate the mixture weight $\hat{\pi}_c$, mean $\hat{\mu}_c$, and covariance $\hat{\Sigma}_c$ of each component. **Compared with a simple weighted average, these statistics provide a richer representation by capturing not only the central tendency, but also the prevalence and intra-component variability of each tissue pattern.** We have added an explanation in the revision.
>
> [1] Song, A.H., et al. "Morphological prototyping for unsupervised slide representation learning in computational pathology." CVPR, 2024.
>
> > KQ2: Relationship between Eq. 2 and Eq. 4?
>
> **Eq. 2 and Eq. 4 describe the same slide decomposition at two levels of explicitness**. Eq. 2 states that the WSI is decomposed into C patch-prototype-aligned components. Eq. 4 makes this concrete by writing the slide representation as the concatenation of estimated GMM statistics $(\hat{\pi}_c,\hat{\mu}_c,\hat{\Sigma}_c)$ for each component. We believe the reviewer may have confused the GMM statistics in Eq. 4 with the later GRFN parameters. **Here, $\hat{\mu}_c$ (bold) and $\hat{\Sigma}_c$ (uppercase) denote the mean vector and covariance matrix of the $c$-th GMM component in feature space, not the $\mu$ and $\sigma$ used later in the GRFN/evidence space**. We have revised the notation to avoid this confusion.
>
> > KQ3: Why does $h_c$ appear in different forms?
>
> **In Eq. 3, $\mathbf{h}_c$ (bold) denotes the patch prototype, a $d$-dimensional vector in $\mathbb{R}^d$. In Sec. 3.4, $h_k$ (non-bold) denotes the fused epistemic precision, which is a scalar.** The subscript $c$ indexes GMM components and $k$ denotes the number of component prototypes within each GMM component, so $h_{c,k}$ is the epistemic precision induced by the $k$-th component prototype for the $c$-th GMM component. We will revise notation to use distinct symbols and avoid this confusion.
>
> > KQ4: Does Eq. 7 create a circular dependency?
>
> The left-hand side represents the fused epistemic precision $h_c$, while the right-hand side represents a collection of individual component prototype precisions $h_{c,k}$. **This equation indicates that the epistemic precision $h_c$ predicted for the c-th GMM component is a weighted aggregation of the K component prototype-level epistemic precisions $(h_{c,k})_{k=1}^K$ within that component.** We have added this explanation in the revision.
>
> > KQ5: How is DPsurv different from EsurvFusion?
>
> DPsurv and EsurvFusion address fundamentally different problem settings. EsurvFusion focuses on multimodal survival fusion across data sources. Its experiments evaluate one image-clinical dataset based on PET/CT radiomic features and three clinical-genomic cancer datasets, **without using histopathology WSIs as input**. By contrast, DPsurv focuses on a single whole-slide image and addresses a different question: how to decompose one WSI into morphology-aware components and perform uncertainty-aware and interpretable survival reasoning within that slide.
>
> In addition, the novelty of DPsurv is not the use of GRFNs alone, but the combination of (i) prototype-guided WSI decomposition into morphology-aware components, (ii) component-wise evidential modeling where each component generates GRFN-based survival evidence, and (iii) evidence-level mixture that aggregates component evidence into interpretable slide-level predictions. We will make this distinction clearer in the revision.
>
> > Response to Limitations discussion
>
> Thank you for clarifying the unclear description.
>
> Firstly, with respect to the application of uncertainty quantification techniques like MC dropout, ensembling, and Bayesian methods, these techniques can indeed be applied to cases other than discrete classification. The distinction we address is with respect to survival analysis. We have revised the description.
>
> Secondly, considering tissue heterogeneity, although the methods of attention-based can capture this information to some extent by paying more attention to the relevant regions, they cannot capture the slide-level compositional information, which is vital in the survival prediction task. Attention-based models have been added in the comparison. We have added attention-based models in comparison and made introduction/related works clearer.

---

> > ### Author Rebuttal · Reviewer_mcnn · 2026-04-03
> >
> > Thank you for the clarifications. The replies properly addressed my concerns.

---

> > > ### Author Response · Authors · 2026-04-04
> > >
> > > Thank you for the positive reassessment and for updating your score. We are grateful that our clarifications have adequately addressed your concerns. Your constructive feedback has been invaluable in strengthening the paper and we sincerely appreciate your willingness to revise your recommendation. We would be happy to incorporate any remaining suggestions.

---

### Official Review · Reviewer_CuW2 · 2026-03-11

**Soundness:** 3
**Presentation:** 4
**Significance:** 3
**Originality:** 3
**Overall Recommendation:** 5
**Confidence:** 3

**Summary:**

The paper present a method for survival prediction from histopathology WSIs, with provided aleatoric and epistemic uncertainty estimates and explainability possibilities. The WSI is patched and encoded by a pre-trained pathology foundation model. The encoded patches are aggregated based on learned prototypes through a Gaussian mixture model. A Gaussian random fuzzy number model is used to provide evidence through component prototypes, enabling estimation of intervals corresponding to epistemic and aleatoric uncertainty of the final survival prediction. It is demonstrated through some examples how the prototypes can be linked to different tissue types, and how a spatial distribution of risk can be visualized. The method is compared on datasets from TCGA against a number of previous methods, both with and without uncertainty-awareness, in terms of prediction performance and calibration quality.

**Compliance With Llm Reviewing Policy:**

Affirmed.

**Final Justification:**

I believe the clarifications and additional experiments will strengthen the paper, and update my recommendation accordingly.

**Key Questions For Authors:**

1. It could be argued that comparison would favor the proposed method if compared methods haven't been tuned for each dataset. Are the compared methods also optimized for each dataset? Is there a separate validation set used for calibration?

2. What is the impact of the selected number of prototypes? There is an ablation for this in the appendix, but how does this impact, e.g., the interpretability perspective? If the number of prototypes is increased, will the same fundamental prototypes be robustly learned? Will the interpretation be made more complicated by having additional prototypes that doesn't encode meaningful concepts?

3. Can the method reliably detect domain shift or out-of-distribution data?

4. In order to have explanations related to different types of tissue, a post-hoc manual annotation needs to be performed. Would it be possible to instead provide annotations beforehand, and train in such way that these align with the prototypes used by the pipeline?

**Limitations:**

* Limitations are not discussed. I would consider it important to do this to give more nuance to the proposed method. For example, is there any risk of introducing some form of bias given the provided interpretation? How robust is the method under different types of domain shifts?

**Strengths And Weaknesses:**

Strengths:
+ Relevant topic that could benefit clinical deployment by providing uncertainty intervals and possibilities for interpretation.

+ Good motivation and formulation of the pipeline, with promising results.

+ Relatively extensive experiments and ablation studies.

Weaknesses:
- The proposed method seems to use per-dataset optimized hyperparameters, while in my understanding the compared methods do not have this benefit.

- The interpretation relies on post-hoc manual annotation of prototypes/concepts provided by pathologists, for each trained model.

- It would be valuable to see some additional evaluation in terms of generalization under domain shift, to verify that the uncertainty estimation and interpretation can reliably detect cases that will provide unreliable prediction.

---

> ### Author Rebuttal · Authors · 2026-03-30
>
> We thank the reviewer for the positive assessment and recognition of the clinical motivation and pipeline design. We address each concern below.
>
> > Response to W1: Fairness of per-dataset hyperparameter selection
>
> Thank you for your comment. **We have added a validation set for K selection, and the details can be seen in our response to Reviewer mLLo (W4).** We report the full updated per-dataset results below (top-5 baselines and DPsurv shown and full table with all 10 methods have been added in the revision). DPsurv is again the top-performing method according to mean C-index, IBS, and IBLL performance over all five cohorts.
>
> | | BRCA | | | BLCA | | | LUAD | | |
> |---|---|---|---|---|---|---|---|---|---|
> | Methods | C-index↑ | IBS↓ | IBLL↓ | C-index↑ | IBS↓ | IBLL↓ | C-index↑ | IBS↓ | IBLL↓ |
> | ABMIL | 0.686±0.06 | 0.752±0.14 | 3.019±1.16 | 0.557±0.04 | 0.535±0.09 | 1.540±0.34 | 0.611±0.11 | 0.627±0.19 | 1.938±0.70 |
> | DSMIL | 0.652±0.04 | 0.694±0.16 | 2.208±0.54 | 0.583±0.03 | 0.417±0.10 | 1.080±0.26 | 0.619±0.10 | 0.472±0.11 | 1.227±0.28 |
> | PANTHER | 0.696±0.05 | 0.837±0.08 | 2.683±0.59 | 0.601±0.06 | 0.530±0.11 | 1.331±0.27 | 0.588±0.04 | 0.667±0.16 | 1.819±0.54 |
> | UMSA | 0.640±0.08 | 0.730±0.11 | 2.336±0.51 | 0.573±0.07 | 0.500±0.10 | 1.381±0.35 | 0.633±0.10 | 0.569±0.17 | 1.663±0.59 |
> | BayesMIL | 0.707±0.05 | 0.721±0.06 | 2.024±0.31 | 0.602±0.06 | 0.414±0.08 | 1.058±0.19 | 0.622±0.11 | 0.461±0.08 | 1.180±0.18 |
> | DPsurv | **0.720±0.03** | **0.199±0.03** | **0.566±0.08** | **0.625±0.06** | **0.410±0.12** | **0.855±0.14** | **0.667±0.08** | **0.381±0.06** | **1.130±0.24** |
>
> | | UCEC | | | KIRC | | | Average | | |
> |---|---|---|---|---|---|---|---|---|---|
> | Methods | C-index↑ | IBS↓ | IBLL↓ | C-index↑ | IBS↓ | IBLL↓ | C-index↑ | IBS↓ | IBLL↓ |
> | ABMIL | 0.636±0.11 | 0.787±0.07 | 2.802±0.17 | 0.703±0.10 | 0.517±0.20 | 1.670±0.94 | 0.639 | 0.644 | 2.194 |
> | DSMIL | 0.690±0.09 | 0.652±0.13 | 1.871±0.45 | 0.693±0.05 | 0.455±0.11 | 1.216±0.28 | 0.647 | 0.538 | 1.520 |
> | PANTHER | 0.709±0.06 | 0.866±0.10 | 3.045±0.86 | 0.683±0.09 | 0.701±0.15 | 1.926±0.48 | 0.655 | 0.720 | 2.161 |
> | UMSA | 0.641±0.11 | 0.758±0.14 | 2.717±1.08 | 0.693±0.08 | 0.506±0.17 | 1.524±0.57 | 0.636 | 0.613 | 1.924 |
> | BayesMIL | 0.736±0.10 | 0.679±0.12 | 1.941±0.47 | 0.703±0.05 | 0.433±0.15 | 1.171±0.41 | 0.674 | 0.542 | 1.475 |
> | DPsurv | **0.766±0.05** | **0.251±0.08** | **0.692±0.19** | **0.741±0.08** | **0.310±0.11** | **0.879±0.28** | **0.704** | **0.310** | **0.824** |
>
> > Response to W2: Role of post-hoc annotation in interpretation
>
> **Post-hoc annotation is only needed for validation and interpretation of the semantic meaning of the learned prototypes and is not a requirement for interpretability of the model.** A logical next step is semi-supervised prototype learning, where partial annotations of tissues are provided to regularize the formation of the prototypes such that some of them correspond to pre-defined concepts of tissues. The trade-off is that this may improve the semantic alignment but could also impair the ability of the model to find new prognostic patterns. We have clarified this in the revision.
>
> > Response to W3: Scope of generalization evaluation
>
> Thank you for giving a new direction. Our current study appears to be showing robustness with moderate cross-site variation (site-stratified TCGA protocol [1]). We will elaborate on this point and add OOD experiments in the revision.
>
> [1] Howard, F.M., et al. "The impact of site-specific digital histology signatures on deep learning model accuracy and bias." Nature Communications 12.1 (2021): 4423.
>
> > Response to KQ1: Are baselines optimized under the same protocol?
>
> In the original setup, **all baselines followed configurations reported as optimal in their papers**. The revised experiment setup (see Reviewer mLLo W4) additionally provides a data-driven K selection rule under the same early-stopping framework. We do not use a separate validation set for calibration. We directly evaluate calibration on the test set via IBS, IBLL, and calibration plots.
>
> > Response to KQ2: How does the number of prototypes affect interpretability?
>
> Thanks for the question. For patch prototypes, the value of C is set to 16 as suggested by PANTHER [2]. In terms of interpretability, it should be high enough to cover important tissue subtypes, yet not so high that the prototypes become redundant and hard to interpret. For component prototypes, although the interpretability might be improved with a high K, it might also become complicated if the prototypes are not significant. we have added a discussion in the revision.
>
> [2] Song, A.H., et al. "Morphological prototyping for unsupervised slide representation learning in computational pathology." CVPR, 2024.
>
> > Response to KQ3–KQ4
>
> KQ3 addressed in W3. KQ4 addressed in W2.
>
> > Response to Limitations discussion
>
> We will discuss the potential bias in interpretations and robustness under different domain-shift types in the revision.

---

> > ### Author Rebuttal · Reviewer_CuW2 · 2026-04-01
> >
> > Thank you for the detailed response and additional experiments. I believe the clarifications and experiments will strengthen the paper, and update my recommendation accordingly.

---

> > > ### Author Response · Authors · 2026-04-02
> > >
> > > Thank you for the positive update and for recognizing the value of our additional experiments. We are pleased that the clarifications and new results have addressed your concerns, and we sincerely appreciate your willingness to update your recommendation. We would be glad to incorporate any remaining suggestions in the final version.

---

### Official Review · Reviewer_mLLo · 2026-03-24

**Soundness:** 3
**Presentation:** 3
**Significance:** 2
**Originality:** 2
**Overall Recommendation:** 3
**Confidence:** 4

**Summary:**

DPsurv tackles the problem of survival prediction from whole-slide images by combining GMMs and GRFNs. The framework produces survival predictions capturing both aleatoric and epistemic uncertainty, while enabling multi-level transparency and explicit uncertainty quantification through a patch prototype assignment map, component prototype evidence reasoning, and relative risk visualization.

**Compliance With Llm Reviewing Policy:**

Affirmed.

**Final Justification:**

I appreciate the authors’ effort in responding to my comments. They have thoroughly addressed my concerns.

In light of these improvements, I have updated my overall recommendation from 3 (Weak Reject) to 4 (Weak Accept).

**Key Questions For Authors:**

Per-WSI GMM does not guarantee component correspondence across slides. The GMM is initialized with global K-means centroids but is then optimized independently per WSI via EM. After adaptation, there is no guarantee that component c in WSI i corresponds to the same morphological concept as component c in WSI j. The downstream component prototypes p_{c,k} are trained assuming this correspondence. Could you please provide an analysis or proof to verify this assumption?

**Limitations:**

Failure case: In Table 7, case TCGA-XF-AAMJ receives a relative risk visualization score of 4, yet the paper provides no analysis or discussion of this failure case. Reporting only the average while ignoring this outlier is misleading.

**Strengths And Weaknesses:**

Strengths:
1. The paper clearly identifies two underexplored problems in WSI survival analysis, namely interpretability and uncertainty quantification, and addresses both in a clinically meaningful way.
2. DPsurv achieves substantially lower IBS and IBLL compared to all baselines, suggesting that the evidential modeling genuinely improves prediction reliability beyond just discrimination.

Weakness:
1. Ablation table:
CP alone collapses C-index (0.663 → 0.538). It suggests CP and EM are not independently beneficial and that module interactions are poorly understood.
2. Computational cost:
The paper claims "relatively small" computational overhead. However, DPsurv (550s/972s) is approximately 1.9×–2.6× slower than PANTHER (292s/381s), its closest conceptual baseline. Especially given that PANTHER is competitive or superior on BRCA.
3. Novelty limited:
The Deep Slide Component Embedding stage (Section 3.3, Eq. 2–5) is mathematically identical to PANTHER (Song et al., 2024). Equation 10 is precisely a weighted average with weights πˆ_c, which is also established in PANTHER.
4. K varies across folds:
For BLCA (K=1–4) and LUAD (K=2–3), different folds of the same dataset use different numbers of component prototypes, making the reported mean C-index across folds difficult to interpret and the cross-fold variance potentially inflated by architectural differences rather than data variance alone.

---

> ### Author Rebuttal · Authors · 2026-03-30
>
> We thank the reviewer for the careful reading and recognition of the paper's strengths. We address the concerns regarding (i) the CP/EM ablation, (ii) computational overhead, (iii) the relationship to PANTHER, and (iv) the model-selection protocol.
>
> > Response to W1: Clarifying the role of CP and EM in the ablation
>
> Thanks for your detailed observation. We have added in the ablation study that CP and EM must work together, as they are not independently beneficial for the following reasons:
>
> CP breaks down the slide-level prediction into component-level evidence. **However, in the absence of EM, these components are uniformly aggregated without reflecting their actual presence in the slide. This can cause noise in performance, particularly for components that are not present in the slide or have a low prevalence.** EM solves this by utilizing the prevalence prior $\pi_c$, which re-weights each component-level evidence according to the tissue composition in each slide. Thus, the performance of CP actually depends on EM for aggregation, which explains why the C-index drops in the case of CP.
>
> The per-dataset CP-without-EM ablation shows consistent degradation across all five cohorts, confirming a structural rather than dataset-specific issue. The per-dataset table is in our response to Reviewer 8mzg (W1).
>
> > Response to W2: Computational overhead relative to PANTHER
>
> We acknowledge the imprecise expression. We have updated the wording of "relatively small" to "acceptable" to match reality.
>
> First, we'd like to address **the efficiency is not a key aspect of DPsurv**. Our purpose is not to be computationally efficient, but to achieve uncertainty-aware and interpretable survival prediction.
>
> Second, we'd like to stress that **DPsurv is still more efficient than classical MIL methods**, such as TransMIL, ABMIL, and BayesMIL, and provides uncertainty quantification and "end-to-end" interpretability on top of that.
>
> > Response to W3: Relationship between Deep Slide Component Embedding and PANTHER
>
> We indeed use PANTHER for slide representation and it is the first step towards end-to-end interpretability (feature-level interpretability, Figure 2A). However, Equation 10 is not established in PANTHER but only share the same parameters $\pi_c$.
>
> The novelty of DPsurv is not a new slide encoder, but a new evidential survival layer: (i) each GMM component is mapped to GRFN-based survival evidence through similarity-weighted component prototypes, and (ii) component evidence is aggregated into uncertainty-aware and interpretable survival prediction via evidence-level mixture. **These complete the remaining steps of interpretability and thus make the interpretability fully end-to-end.**
>
> While Eq. 10 shares the weights $\pi_c$ with PANTHER, **it operates in the evidence space rather than in the feature space.** This is grounded in the theoretical consistency between GMMs and mixture GRFNs (**Appendix E**), following the parametric belief function framework [1]. The shared use of $\pi_c$ is not incidental but reflects a principled theoretical bridge between GMM-based representation and evidential reasoning.
>
> [1] Denœux, T. "Parametric families of continuous belief functions based on generalized Gaussian random fuzzy numbers." Fuzzy Sets and Systems 471 (2023): 108679.
>
> > Response to W4: Model-selection protocol across folds
>
> Thank for pointing out. K needs to adjust based on different training samples for splits to better capture potential component subtypes. **We have conducted experiment to address this concern.**
>
> Within each fold, 15% of training data is held out as validation set. DPsurv uses a unified learning rate of 1e-4. K is selected from {1,2,3,4} by applying early stopping on validation set. All baselines are retrained under the same validation/early-stopping protocol using their originally reported optimal configurations.
>
> **DPsurv achieves an average C-index of 0.704, IBS of 0.310, and IBLL of 0.824**, which is the best across all three metrics among all compared methods. The full per-dataset table is in our response to Reviewer CuW2 (W1).
>
> > Response to Key Question: Cross-slide component alignment
>
> We are pleased to provide an analysis. All WSIs utilize the cohort-level patch prototypes as an initialization and the EM for each WSI only makes some small modifications around this shared anchor. **This is also supported from an empirical perspective**, where the cross-slide alignment is stable, as is also the case in PANTHER [2], as well as the high coherence from the pathologist evaluation (Appendix I / Table 7).
>
> [2] Song, A.H., et al. "Morphological prototyping for unsupervised slide representation learning in computational pathology." CVPR, 2024.
>
> > Response to Limitations
>
> We have added TCGA-XF-AAMJ as a targeted failure analysis. This case demonstrates that when the tumor component is small, evidence of risk is significantly discounted, increasing uncertainty and leading to challenges in risk ranking.

---

> > ### Author Rebuttal · Reviewer_mLLo · 2026-04-03
> >
> > I thank the authors for their response and maintain my original score.

---

> > > ### Author Response · Authors · 2026-04-04
> > >
> > > Thank you for your acknowledgement. We noticed that you selected "(b) Partially resolved - I have follow-up questions," but we did not find specific follow-up questions in your response. Could you kindly clarify if there are remaining concerns we can address? We would be happy to provide further clarification.

---

### Decision · Program_Chairs · 2026-04-30

**Decision:**

Accept (regular)

**Comment:**

This paper presents DPsurv, a dual-prototype evidential fusion framework for whole slide image survival prediction that provides uncertainty-aware prediction together with interpretable survival reasoning. Reviewers agree that the problem is important and that the paper shows good empirical results, especially on calibration-related metrics, while also offering meaningful uncertainty estimation and multi-level interpretability. The main concerns focus on hyperparameter selection fairness, presentation clarity, method complexity, and limited validation of robustness and clinical generalization. The rebuttal addresses most of these concerns by clarifying the training and selection protocol, improving the explanation of the method, and better positioning the work relative to related approaches, which leads reviewers to view the paper more positively. Some limitations still remain, especially regarding broader robustness validation and the scale of the interpretability study, but they do not seem to outweigh the overall strengths of the work. The authors should incorporate the clarified content in the final version.